# Pretrain then Finetune:
# Getting the Most from Your Labeled Dataset in Tabular Deep Learning

## Abstract

Similarly to computer vision and NLP, deep learning models for tabular data can benefit from pretraining, which is not the case for traditional ML models, such as gradient boosted decision trees (GBDT). Although the pretraining techniques for tabular data are actively studied, the existing works mostly focus on the unsupervised pretraining, implying the access to a large amount of unlabeled data in addition to the labeled target dataset. By contrast, pretraining in the fully supervised setting, when the available data is fully labeled and directly represents the downstream tabular task, receives significantly less attention. Moreover, the existing works on pretraining typically consider only the simplest MLP architectures and do not cover the recently proposed tabular models.

In this work, we aim to identify the best practices for pretraining in tabular DL that can be universally applied to different datasets and architectures in the fully supervised setting. Among our findings, we show that using the object target labels during the pretraining stage is beneficial for the downstream performance and advocate several target-aware pretraining objectives. Overall, our experiments demonstrate that properly performed pretraining significantly increases the performance of tabular DL models on fully supervised problems.

## 1 Introduction

Tabular problems are ubiquitous in industrial ML applications, which include data described by a set of heterogeneous features, such as learning-to-rank, click-through rate prediction, credit scoring, and many others. Despite the current dominance of deep learning models in the ML literature, for tabular problems, the "old-school" decision tree ensembles (e.g., GBDT) are often the top choice for practitioners. Only recently, several works have proposed the deep models that challenge the supremacy of GBDT in the tabular domain (Arik & Pfister, 2020; Gorishniy et al., 2021; 2022) and suggest that the question "tabular DL or GBDT" is yet to be answered.

An important advantage of deep models over GBDT is that they can potentially achieve higher performance via pretraining their parameters with a properly designed objective. These pretrained parameters, then, serve as a better than random initialization for subsequent finetuning for downstream tasks. For computer vision and NLP domains, pretraining is a de facto standard and is shown to be necessary for the state-of-the-art performance (He et al., 2021; Devlin et al., 2019). For tabular problems such a consensus is yet to be achieved as well as the best practices of tabular pretraining are to be established. While a large number of prior works address the pretraining of tabular DL models (Yoon et al., 2020; Bahri et al., 2022; Ucar et al., 2021; Darabi et al., 2021), it is challenging to make reliable conclusions about pretraining efficacy in tabular DL from the literature since experimental setups vary significantly. Moreover, most evaluation protocols assume the unlabeled data is abundant and only use a small subset of labels from each dataset during finetuning for evaluation – demonstrating pretraining efficacy, but somewhat limiting the performance of supervised baselines on these datasets. Such protocols of "unsupervised pretraining" are common in vision or NLP domains, where huge "extra" data is available on the Internet.

In contrast, in our work, we focus on the setup with fully labeled tabular datasets to understand if pretraining helps tabular DL in a fully supervised setting and compare pretraining methods to

the strong supervised baselines. We argue that this setup is reasonable in practice, since for many tabular problems there is no publicly available relevant data to pretrain on. In this setup, we perform a systematic experimental evaluation of several pretraining objectives, identify the superior ones, and describe the practical details of how to perform tabular pretraining optimally. Our main findings, which are important for practitioners, are summarized below:

- Pretraining provides substantial gains over well-tuned supervised baselines in the fully supervised setup.

- Simple self-prediction based pretraining objectives are comparable to the objective based on contrastive learning. To the best of our knowledge, this behavior was not reported before in tabular DL.

- The object labels can be exploited for more effective pretraining. In particular, we describe several "target-aware" objectives and demonstrate that they often outperform their "unsupervised" counterparts.

- The pretraining provides the most noticeable improvements for the vanilla MLP architecture. In particular, their performance after pretraining becomes comparable to the state-of-the-art models trained from scratch, which is important for practitioners, who are interested in simple and efficient solutions.

- The ensembling of pretrained models is beneficial. It indicates that the pretraining stage does not significantly decrease the diversity of the models, despite the fact that all the models are initialized by the same set of parameters.

Overall, our work provides a set of recipes for practitioners interested in tabular pretraining, which results in higher performance for most of the tasks. The code of our experiments is available online.[1]

## 2 RELATED WORK

Here we briefly review the lines of research that are relevant to our study.

**Status quo in tabular DL.** A plethora of recent works have proposed a large number of deep models for tabular data (Klambauer et al., 2017; Popov et al., 2020; Arik & Pfister, 2020; Song et al., 2019; Wang et al., 2017; Badirli et al., 2020; Hazimeh et al., 2020; Huang et al., 2020; Gorishniy et al., 2021; Kossen et al., 2021). Several systematic studies, however, reveal that these models typically do not consistently outperform the decision tree ensembles, such as GBDT (Gradient Boosting Decision Tree) (Chen & Guestrin, 2016; Prokhorenkova et al., 2018; Ke et al., 2017), which are typically the top-choice in various ML competitions (Gorishniy et al., 2021; Shwartz-Ziv & Armon, 2021). Additionally, several works have shown that the existing sophisticated architectures are not consistently superior to properly tuned simple models, such as MLP and ResNet (Gorishniy et al., 2021; Kadra et al., 2021). Finally, the recent work (Gorishniy et al., 2022) has highlighted that the appropriate embeddings of numerical features in the high-dimensional space are universally beneficial for different architectures. In our work, we experiment with pretraining of both traditional MLP-like models and advanced embedding-based models proposed in Gorishniy et al. (2022).

**Pretraining in deep learning.** For domains with structured data, like natural images or texts, pretraining is currently an established stage in the typical pipelines, which leads to higher general performance and better model robustness (He et al., 2021; Devlin et al., 2019). Pretraining with the auto-encoding objective was also previously studied as a regularization strategy helping in the optimization process without large scale pretraining datasets (Erhan et al., 2010; El-Nouby et al., 2021; Krishna et al., 2022). During the last years, several families of successful pretraining methods have been developed. An impactful line of research on pretraining is based on the paradigm of contrastive learning, which effectively enforces the invariance of the learned representations to the human-specified augmentations (Chen et al., 2020; He et al., 2020). Another line of methods exploits the idea of self-prediction, i.e., these methods require the model to predict certain parts of the input given the remaining parts (He et al., 2021; Devlin et al., 2019). In the vision community, the self-prediction based methods are shown to be superior to the methods that use contrastive learning

---

[1]Code: https://anonymous.4open.science/r/pretrains-DD02

objectives (He et al., 2021). In our experiments, we demonstrate that self-prediction based objectives are comparable to the contrastive learning ones on tabular data, while being much simpler.

**Pretraining for the tabular domain.** Numerous pretraining methods were recently proposed in several recent works on tabular DL (Arik & Pfister, 2020; Yoon et al., 2020; Darabi et al., 2021; Ucar et al., 2021; Somepalli et al., 2021; Kossen et al., 2021; Padhi et al., 2021). However, most of these works do not focus on the pretraining objective per se and typically introduce it as a component of their tabular DL pipeline. Moreover, the experimental setup varies significantly between methods. Therefore, it is difficult to extract conclusive evidence about pretraining effectiveness from the literature. To the best of our knowledge, there is only one systematic study on the tabular pretraining (Bahri et al., 2022), but its experimental evaluation is performed only with the simplest MLP models, and we found that the superiority of the contrastive pretraining, reported in Bahri et al. (2022), does not hold for tuned models in our setup, where contrastive objective is comparable to the simpler self-prediction objectives.

# 3 REVISITING PRETRAINING OBJECTIVES

In this section, we evaluate the typical pretraining objectives under the unified experimental setup on the number of datasets from the literature on tabular DL. Our goal is to answer whether pretraining generally provides significant benefits over tuned models trained from scratch and to identify the pretraining objectives that lead to the best downstream task performance.

## 3.1 EXPERIMENTAL SETUP

We mostly follow the experimental setup from Gorishniy et al. (2021) and describe its main details here for completeness.

**Notation.** Each tabular dataset $(\mathbb{X}, \mathbb{Y})$ is represented by a set of pairs $\{(x_i, y_i)\}_{i=1}^n$, where $x_i = (x_i^1, \ldots, x_i^m) \in \mathbb{X}$ are the object's features (both numerical and categorical) and $y_i \in \mathbb{Y}$ is the target variable. The downstream task is either regression $\mathbb{Y} = \mathbb{R}$ or classification $\mathbb{Y} = \{1, \ldots, k\}$. Each model has the backbone $f(x|\theta)$ that is followed by two separate heads: a pretraining head $h(z|\mu)$ and a downstream task head $g(z|\lambda)$, with learnable parameters $\theta, \mu, \lambda$ respectively, and $z = f(x|\theta)$ denotes the output of the backbone for an input object $x$.

**Datasets.** We evaluate the pretraining methods on a curated set of eleven middle to large scale datasets used in prior literature on tabular deep learning. The benchmark is biased towards tasks, where tuned MLP models were shown to be inferior to GBDT (Gorishniy et al., 2021) since we aim to understand if pretraining can help the deep models to beat the "shallow" ones. The datasets represent a diverse set of tabular data problems with classification and regression targets. The main dataset properties are summarized in Table 1.

We report ROC-AUC for all binary classification datasets, accuracy for multi-class classification datasets and RMSE for regression datasets, with OT being the one exception, where we report log-loss, as it was used as a default metric in the corresponding Kaggle competition. We use the quantile-transform from the Scikit-learn library (Pedregosa et al., 2011) to preprocess the numerical features for all datasets except OT, where the absence of such transformation was shown to be superior (Gorishniy et al., 2022). Additional information about the datasets is provided in Appendix A.

|  | GE | CH | CA | HO | OT | HI | FB | AD | WE | CO | MI |
|---|---|---|---|---|---|---|---|---|---|---|---|
| #objects | 9873 | 10000 | 20640 | 22784 | 61878 | 98049 | 197080 | 48842 | 397099 | 581012 | 1200192 |
| #num.features | 32 | 10 | 8 | 16 | 93 | 28 | 50 | 6 | 123 | 54 | 136 |
| #cat.features | 0 | 1 | 0 | 0 | 0 | 0 | 1 | 8 | 0 | 0 | 0 |
| metric | Acc. | AUC | RMSE | RMSE | Log. Loss | AUC | RMSE | AUC | RMSE | Acc. | RMSE |
| #classes | 5 | 2 | – | – | 9 | 2 | – | 2 | – | 7 | – |

Table 1: Datasets used for the experiments.

**Models.** We use MLP as a simple deep baseline to compare and ablate the methods. Our implementation of MLP exactly follows Gorishniy et al. (2021), the model is regularized by dropout and weight decay. As more advanced deep models, we evaluate MLP equipped with numerical feature embeddings, specifically, target-aware piecewise linear encoding (MLP-T-LR) and embeddings with periodic activations (MLP-PLR) from Gorishniy et al. (2022). These models represent the current state-of-the-art solution for tabular DL (Gorishniy et al., 2022), and are of interest as most prior work on pretraining in tabular DL focus on pretraining with the simplest MLP models in evaluation. The implementation of models with numerical embeddings follows Gorishniy et al. (2022). We use AdamW (Loshchilov & Hutter, 2019) optimizer, do not use learning rate schedules and fix batch sizes for each dataset based on the dataset size (see Appendix A).

**Pretraining** is always performed directly on the target dataset and does not exploit additional data. The learning process thus comprises two stages. On the first stage, the model parameters are optimized w.r.t. the pretraining objective. On the second stage, the model is initialized with the pretrained weights and finetuned on the downstream classification or regression task. We focus on the fully-supervised setup, i.e., assume that target labels are provided for all dataset objects. Typically, pretraining stage involves the input corruption: for instance, to generate positive pairs in contrastive-like objectives or to corrupt the input for reconstruction in self-prediction based objectives. We use random feature resampling as a proven simple baseline for input corruption in tabular data (Bahri et al., 2022; Yoon et al., 2020), we tune the corruption probability in all experiments. Learning rate and weight decay are shared between the two stages (see Table 15 for the ablation). We fix the maximum number of pretraining iterations for each dataset at $100k$. On every $10k$-th iteration, we compute the value of the pretraining loss using the hold-out validation objects for early-stopping on large-scale WE, CO and MI datasets. On other datasets we directly finetune the current model every $10k$-th iteration and perform early-stopping based on the target metric after finetuning (we do not observe much difference between early stopping by loss or by target metric, see Table 16).

**Hyperparameters & Evaluation.** Hyperparameter tuning is crucial for a fair comparison, therefore, we use Optuna (Akiba et al., 2019) to optimize the model and pretraining hyperparameters for each method on each dataset. We use the validation subset of each dataset for hyperparameter tuning. The exact search spaces for the hyperparameters of each method are provided in Appendix B.

We run the tuned configuration of each pretraining method with 15 random seeds and report the average metric on the test splits. When comparing to GBDT, we obtain three ensembles by splitting the fifteen single model predictions into three disjoint subsets of five models and averaging predictions within each subset. Then, we report the average metric over the three ensembles.

## 3.2 COMPARING PRETRAINING OBJECTIVES

Here we compare the contrastive learning and self-prediction objectives from prior work in the described setup. For contrastive learning, we follow the method described in Bahri et al. (2022): use InfoNCE loss, consider corrupted inputs $\hat{x}$ as positives for $x$ and the rest of the batch as negatives. For self-prediction methods, we evaluate two objectives: the first one is the reconstruction of the original $x$, given the corrupted input $\hat{x}$ (the reconstruction loss is computed for all columns), the second one is the binary mask prediction, where the objective is to predict the mask vector $m$ indicating the corrupted columns from the corrupted input $\hat{x}$. The results of the comparison are in Table 2. We summarize our key findings below.

**Contrastive is not superior.** Both the reconstruction and the mask prediction objectives are preferable to the contrastive objective. The two self-prediction objectives have the advantage of being conceptually simpler, and easier to implement, while also being less resource-intensive (no need for the second view of augmented examples in each batch, simpler loss function). We thus recommend the self-prediction based objectives as a practical solution for pretraining in tabular DL.

**Pretraining is beneficial for the state-of-the-art models.** Models with the numerical feature embeddings also benefit from pretraining with either reconstruction or mask prediction demonstrating the top performance on the downstream task. However, the improvement is typically less noticeable compared to the vanilla MLPs.

**There is no universal solution between self-prediction objectives.** We observe that for some datasets the reconstruction objective outperforms the mask prediction (OT, WE, CO, MI), while

| | GE ↑ | CH ↑ | CA ↓ | HO ↓ | OT ↓ | HI ↑ | FB ↓ | AD ↑ | WE ↓ | CO ↑ | MI ↓ |
|---|---|---|---|---|---|---|---|---|---|---|---|
| | | | | | MLP | | | | | | |
| no pretraining | 0.635 | 0.849 | 0.506 | 3.156 | 0.479 | 0.801 | 5.737 | 0.908 | 1.909 | 0.963 | 0.749 |
| contrastive | 0.672 | **0.855** | 0.455 | **3.056** | 0.469 | **0.813** | 5.697 | 0.910 | 1.881 | 0.960 | 0.748 |
| rec | 0.662 | 0.853 | **0.445** | **3.044** | **0.466** | 0.805 | **5.641** | **0.910** | **1.875** | **0.965** | **0.746** |
| mask | **0.691** | **0.857** | 0.454 | 3.113 | 0.472 | **0.814** | **5.681** | **0.912** | 1.883 | 0.964 | 0.748 |
| | | | | | MLP-PLR | | | | | | |
| no pretraining | 0.668 | **0.858** | 0.469 | **3.008** | 0.483 | 0.809 | **5.608** | 0.926 | 1.890 | 0.969 | 0.746 |
| rec | 0.667 | 0.852 | **0.439** | 3.031 | **0.472** | 0.808 | **5.571** | 0.926 | **1.877** | **0.971** | **0.745** |
| mask | **0.685** | **0.863** | **0.434** | **3.007** | 0.477 | **0.818** | **5.586** | **0.927** | 1.911 | **0.970** | 0.748 |
| | | | | | MLP-T-LR | | | | | | |
| no pretraining | 0.634 | **0.866** | 0.444 | 3.113 | 0.482 | 0.805 | 5.520 | 0.925 | 1.897 | 0.968 | 0.749 |
| rec | **0.652** | 0.857 | **0.424** | 3.109 | **0.472** | 0.808 | **5.363** | 0.924 | **1.861** | **0.969** | **0.746** |
| mask | **0.654** | **0.868** | **0.424** | **3.045** | **0.472** | **0.818** | 5.544 | **0.926** | 1.916 | **0.969** | 0.748 |

Table 2: Results for pretraining deep models with different objectives. We report metrics averaged over 15 seeds, bold entries correspond to results that are statistically significantly better (we use Tukey HSD test). The comparisons are separate for different models. ↑ corresponds to accuracy and ROC-AUC metrics, ↓ corresponds to RMSE and log-loss for OT. "no pretraining" stands for the supervised baseline, initialized with random weights.

on others the mask prediction is better (GE, CH, HI, AD). We also note that the mask prediction objective sometimes leads to unexpected performance drops for models with numerical embeddings (WE, MI), we do not observe significant performance drops for the reconstruction objective.

**The main takeaway:** simple pretraining strategies based on self-prediction lead to significant improvements in the downstream accuracy compared to the tuned supervised baselines learned from scratch across different tabular DL models and datasets. In practice, we recommend trying both reconstruction and mask prediction as tabular pretraining baselines, as either one might show superior performance depending on the dataset being used.

## 4 TARGET-AWARE PRETRAINING OBJECTIVES

In this section, we show that exploiting the target variables during the pretraining stage can further increase the downstream performance. Specifically, we evaluate several strategies to leverage information about targets during pretraining, identify the best ones and compare them to GBDT. Below we describe a list of target-aware pretraining objectives that we investigate.

**Supervised loss with augmentations**. A straightforward way to incorporate the target variable into the pretraining is by using the input corruption as an augmentation for the standard supervised learning objective. An important difference of this baseline in our setup to the one in Bahri et al. (2022) is that we treat learning on corrupted samples as a pretraining stage and finetune the entire model on the full uncorrupted dataset afterwards (we ablate this in subsection 5.3).

**Supervised loss with augmentations + self-prediction**. We evaluate a natural extension to the above baseline: a combination of the supervised objective with the unsupervised self-prediction. Note, that during the pretraining stage both losses are calculated on corrupted inputs, while the finetuning is performed on the non-corrupted dataset. For the self-prediction objectives we evaluate both the reconstruction and the mask prediction. We use different prediction heads for supervised and self-prediction objectives. We sum supervised and self-prediction losses with equal weights.

**Target-aware pretraining**. An alternative to the approaches described above is the modification of the pretraining task itself. An example of this approach is supervised contrastive learning (Khosla et al., 2020), where the target variable is used to sample positive and negative examples. We introduce the target variable into the self-prediction based objectives with two modifications.

First, we condition the mask prediction or the reconstruction head on the original input's target by concatenating the hidden representation from the backbone network $z = f(\hat{x})$ with the target variable representation before passing it to the pretraining head to obtain predictions $p =$

| | GE↑ | CH↑ | CA↓ | HO↓ | OT↓ | HI↑ | FB↓ | AD↑ | WE↓ | CO↑ | MI↓ | Avg. Rank |
|---|---|---|---|---|---|---|---|---|---|---|---|---|
| | | | | | MLP | | | | | | | |
| no pretraining | 0.635 | 0.849 | 0.506 | 3.156 | 0.479 | 0.801 | 5.737 | 0.908 | 1.909 | 0.963 | 0.749 | $5.5 \pm 1.4$ |
| mask | 0.691 | 0.857 | 0.454 | 3.113 | 0.472 | 0.814 | 5.681 | 0.912 | 1.883 | 0.964 | 0.748 | $3.8 \pm 1.4$ |
| rec | 0.662 | 0.853 | 0.445 | **3.044** | 0.466 | 0.805 | 5.641 | 0.910 | 1.875 | 0.965 | 0.746 | $3.6 \pm 1.5$ |
| sup | 0.693 | 0.856 | 0.441 | 3.077 | **0.459** | 0.814 | 5.689 | 0.914 | 1.883 | 0.968 | 0.748 | $3.0 \pm 1.0$ |
| mask + target | 0.683 | 0.857 | 0.434 | **3.056** | 0.468 | **0.819** | 5.633 | 0.914 | 1.876 | 0.965 | 0.748 | $2.9 \pm 1.3$ |
| rec + target | 0.659 | 0.853 | 0.454 | **3.044** | 0.463 | 0.806 | 5.636 | 0.909 | 1.884 | 0.965 | **0.745** | $3.7 \pm 1.9$ |
| mask + sup | 0.693 | 0.857 | 0.436 | 3.099 | **0.458** | 0.817 | 5.685 | 0.915 | 1.873 | 0.967 | 0.748 | $2.7 \pm 1.2$ |
| rec + sup | 0.684 | 0.854 | 0.436 | **3.012** | **0.456** | 0.815 | 5.672 | 0.911 | **1.862** | 0.967 | 0.747 | $2.6 \pm 1.5$ |
| | | | | | MLP-PLR | | | | | | | |
| no pretraining | 0.668 | 0.858 | 0.469 | **3.008** | 0.483 | 0.809 | 5.608 | 0.926 | 1.890 | 0.969 | 0.746 | $3.5 \pm 1.7$ |
| mask | 0.685 | **0.863** | 0.434 | **3.007** | 0.477 | 0.818 | 5.586 | 0.927 | 1.911 | 0.970 | 0.748 | $2.8 \pm 1.7$ |
| rec | 0.667 | 0.852 | 0.439 | **3.031** | 0.472 | 0.808 | 5.571 | 0.926 | 1.877 | **0.971** | **0.745** | $2.6 \pm 1.2$ |
| sup | **0.710** | 0.859 | 0.433 | 3.136 | 0.479 | 0.811 | 5.521 | 0.924 | 1.873 | **0.971** | 0.748 | $2.5 \pm 1.2$ |
| mask + target | 0.694 | 0.862 | **0.425** | 3.023 | 0.474 | **0.821** | 5.537 | **0.929** | 1.911 | 0.969 | 0.749 | $2.5 \pm 1.9$ |
| rec + target | 0.688 | 0.860 | 0.445 | 3.064 | 0.475 | 0.812 | 5.507 | 0.927 | 1.887 | **0.971** | 0.748 | $2.7 \pm 1.3$ |
| mask + sup | **0.711** | **0.866** | 0.441 | 3.129 | 0.480 | 0.813 | 5.480 | 0.925 | 1.875 | 0.969 | **0.745** | $2.5 \pm 1.4$ |
| rec + sup | **0.709** | 0.858 | 0.433 | **3.059** | 0.465 | 0.807 | 5.571 | 0.927 | **1.865** | **0.971** | **0.745** | $1.9 \pm 1.2$ |
| | | | | | MLP-T-LR | | | | | | | |
| no pretraining | 0.634 | **0.866** | 0.444 | 3.113 | 0.482 | 0.805 | 5.520 | 0.925 | 1.897 | 0.968 | 0.749 | $3.9 \pm 1.7$ |
| mask | 0.654 | **0.868** | **0.424** | **3.045** | 0.472 | 0.818 | 5.544 | 0.926 | 1.916 | 0.969 | 0.748 | $2.8 \pm 1.7$ |
| rec | 0.652 | 0.857 | **0.424** | 3.109 | 0.472 | 0.808 | **5.363** | 0.924 | **1.861** | 0.969 | **0.746** | $2.5 \pm 1.4$ |
| sup | 0.682 | 0.860 | 0.430 | 3.135 | 0.471 | 0.807 | 5.525 | 0.927 | 1.893 | **0.971** | 0.747 | $2.8 \pm 1.5$ |
| mask + target | 0.649 | **0.865** | **0.421** | **3.058** | 0.474 | **0.820** | 5.644 | **0.929** | 1.924 | 0.969 | 0.749 | $2.8 \pm 2.1$ |
| rec + target | 0.668 | **0.864** | 0.440 | 3.113 | 0.473 | 0.806 | 5.493 | 0.927 | **1.862** | 0.969 | **0.746** | $2.5 \pm 1.4$ |
| mask + sup | 0.676 | 0.858 | 0.429 | 3.199 | 0.468 | 0.814 | 5.510 | 0.926 | 1.869 | **0.971** | 0.748 | $2.5 \pm 0.8$ |
| rec + sup | 0.678 | **0.865** | 0.437 | 3.112 | 0.462 | 0.807 | 5.516 | 0.927 | **1.862** | **0.970** | 0.748 | $2.4 \pm 1.2$ |

Table 3: Variations of the target-aware pretraining schemes. Notation follows Table 2. Bold results indicate statistically significant winners across all models and methods. "+ target" denotes target-conditioned pretraining, "+ sup" denotes auxiliary supervised head.

$h(\texttt{concat}[z, y])$. For classification datasets we encode $y$ with one-hot-encoding, for regression targets we use the standard scaling.

Second, we change the input corruption scheme by sampling the replacement from the feature target conditional distribution where a target is different to the original. Intuitively, corrupting the input object $x$ in the direction of the target different to the original makes the pretraining task more correlated with the downstream target prediction. Concretely, given an object-target pair $(x_i, y_i)$, we sample a new target $\hat{y}_i$ from a uniform distribution over the set $\{y \mid y \neq y_i\}$ [2], then each feature $x^j$ is replaced with a sample from the $p(x^j | \hat{y}_i)$ distribution, instead of $p(x^j)$.

### 4.1 COMPARING TARGET-AWARE OBJECTIVES

Here we compare the strategies of incorporating the target variable into pretraining. The results of the comparison are in Table 3. Our key findings are formulated below.

**Supervised loss with augmentations is another strong baseline for MLP**. Pretraining with the supervised loss on corrupted data consistently improves over supervised training from scratch for the MLP. This objective is a strong baseline along with the self-prediction based objectives. However, for models with numerical embeddings the supervised objective with corruptions is less consistent and sometimes is inferior to training from scratch, thus for these models we recommend the self-prediction objectives alone as baselines.

---

[2] For regression problems, when the target variable is continuous, we preliminarily discretize it into $n$ uniform bins, where $n$ is chosen according to the Freedman–Diaconis rule (Freedman & Diaconis, 1981).

| | GE ↑ | CH ↑ | CA ↓ | HO ↓ | OT ↓ | HI ↑ | FB ↓ | AD ↑ | WE ↓ | CO ↑ | MI ↓ | Avg. Rank |
|---|---|---|---|---|---|---|---|---|---|---|---|---|
| CatBoost | 0.692 | **0.864** | 0.430 | 3.093 | 0.450 | 0.807 | 5.226 | 0.928 | 1.801 | 0.967 | **0.741** | 2.6 ± 1.4 |
| XGBoost | 0.683 | 0.860 | 0.434 | 3.152 | 0.454 | 0.805 | 5.338 | 0.927 | **1.782** | 0.969 | **0.742** | 3.0 ± 1.5 |
| MLP | | | | | | | | | | | | |
| no pretraining | 0.656 | 0.852 | 0.482 | 3.055 | 0.467 | 0.805 | 5.666 | 0.910 | 1.850 | 0.968 | 0.747 | 4.8 ± 1.1 |
| mask + target | 0.709 | 0.860 | 0.414 | 2.949 | 0.457 | **0.828** | 5.551 | 0.916 | 1.809 | 0.969 | 0.746 | 2.8 ± 1.2 |
| rec + sup | 0.709 | 0.859 | 0.419 | 2.951 | **0.442** | 0.817 | 5.531 | 0.913 | 1.801 | 0.973 | 0.745 | 2.5 ± 1.2 |
| MLP-PLR | | | | | | | | | | | | |
| no pretraining | 0.695 | **0.864** | 0.454 | 2.953 | 0.470 | 0.814 | 5.324 | 0.928 | 1.835 | **0.974** | 0.744 | 2.6 ± 1.2 |
| mask + target | **0.719** | **0.866** | **0.407** | 2.952 | 0.458 | **0.828** | 5.373 | **0.930** | 1.849 | 0.973 | 0.745 | 2.1 ± 1.2 |
| rec + sup | **0.737** | 0.862 | 0.424 | 2.964 | 0.449 | 0.811 | **5.124** | **0.929** | 1.813 | **0.974** | 0.744 | 2.0 ± 1.0 |
| MLP-T-LR | | | | | | | | | | | | |
| no pretraining | 0.662 | **0.868** | 0.437 | 3.028 | 0.472 | 0.808 | 5.424 | 0.927 | 1.850 | 0.972 | 0.747 | 3.7 ± 1.1 |
| mask + target | 0.673 | **0.868** | 0.410 | **2.894** | 0.460 | **0.827** | 5.458 | **0.930** | 1.849 | 0.972 | 0.746 | 2.4 ± 1.4 |
| rec + sup | 0.705 | **0.866** | 0.425 | 3.057 | **0.444** | 0.814 | 5.422 | 0.927 | 1.811 | **0.974** | 0.746 | 2.5 ± 1.1 |

Table 4: Comparison of pretrained models to GBDT. Notation follows Table 2. Results represent ensembles of models. Bold entries correspond to the overall statistically significant best entries.

**Target-aware objectives demonstrate the best performance**. Both the supervised loss with self-prediction and modified self-prediction objectives improve over the unsupervised pretraining baselines across datasets and model architectures. For the objective with the combination of supervised and self-prediction losses the variation with the reconstruction loss is the most consistent across models and dataset with no performance drops below the pretraining-free baseline. The variant with mask prediction shows similar performance and stability for the MLP, but is not as good as reconstruction for models with numerical embeddings. For the target-aware self-prediction objectives, the modified mask prediction delivers significant improvements over its unsupervised counterpart. Modified reconstruction objective, however, does not improve over its unsupervised counterpart.

**Main takeaways**: Target-aware objectives help further increase the downstream performance, improving upon their unsupervised counterparts. For the reconstruction based self-prediction baseline the addition of the supervised loss is most beneficial ("rec + sup" from Table 4), for the mask prediction objective it's target-aware modification provides more improvements ("mask + target" from Table 4). A simple MLP model pretrained with those "target-aware" objectives often reaches or surpasses complex models with numerical embeddings trained from scratch. In practice, we recommend first trying the baseline pretraining objectives ("rec", "mask", "sup" from Table 4), choosing the suitable baseline for the dataset and improving it accordingly: supervised loss for the reconstruction and target-aware modification for the mask prediction.

## 4.2 COMPARISON TO GBDT

Here we compare MLPs and MLPs with numerical feature embeddings pretrained with the supervised loss with reconstruction and the target-aware mask prediction objectives to the GBDTs. Table 4 shows the results of the comparison.

We observe that both pretraining with target aware objectives and using numerical feature embeddings consistently improve the performance of the simple MLP backbone. In particular, MLP coupled with target aware pretraining starts to outperform GBDT on 4 datasets (GE, CA, OT, HI). Combined with numerical feature embeddings, pretraining improves MLP performance further, making it superior to GBDT on the majority of the datasets, with two exceptions in WE and MI.

## 5 ANALYSIS

### 5.1 INVESTIGATING THE PROPERTIES OF PRETRAINED MODELS

In this section, we provide a possible explanation of why the incorporation of the target variable into pretraining can lead to better downstream task performance. We do this through the experiments on

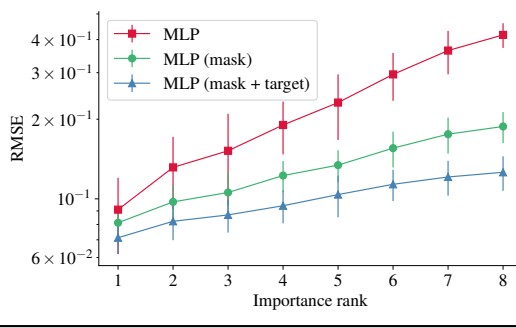

Figure 1: The decodability of object feature from the intermediate representations computed by the pretrained models and the models trained from scratch. The pretrained models decently capture the information about all the features, while the randomly initialized models capture the most informative features and suppress the others.

| | GE↑ | CH↑ | CA↓ | HO↓ | OT↓ | HI↑ | FB↓ | AD↑ | WE↓ | CO↑ | MI↓ |
|---|---|---|---|---|---|---|---|---|---|---|---|
| single | 0.683 | 0.857 | 0.434 | 3.056 | 0.468 | 0.819 | 5.633 | 0.914 | 1.876 | 0.965 | 0.748 |
| standard ensemble | **0.709** | **0.860** | **0.414** | **2.949** | **0.457** | **0.828** | **5.551** | **0.916** | **1.809** | **0.969** | **0.746** |
| efficient ensemble | 0.702 | 0.861 | 0.411 | 2.967 | 0.461 | 0.825 | 5.590 | 0.917 | 1.820 | 0.969 | 0.746 |

Table 5: Efficient ensembling for MLP mask + target. Efficient ensemble consists of models finetuned from one pretrained checkpoint, standard ensemble consists of models from different pretraining runs (random seeds).

the controllable synthetic data. Here we describe the properties and the generation process of the data and our observations on the differences of the pretraining schemes.

We follow the synthetics generation protocol described in Gorishniy et al. (2021) with a modification that allows for the manual control of the feature importance for the particular prediction task. Concretely, we generate the objects features $\{x_i\}_{i=1}^n$ as samples from the multivariate Gaussian distribution with zero mean and covariance matrix $\Sigma$ with identical diagonal and a constant $c = 0.5$ everywhere else. Additionally, we sample a vector $p \in \mathbb{R}^m$ from the Dirichlet distribution $p \sim \mathrm{Dir}(1_m)$ and let $p$ define the influence of the objects features on the target by sampling features for tree splits with probabilities defined by this vector. Intuitively if a particular feature is often used for splitting in the nodes of a decision tree, it would have more influence on the target variable. Indeed, we find the feature importances[3] correlate well with the predefined vector $p$. For more details on the synthetic dataset construction see Appendix C.

For each generated dataset, we then check whether the finetuned models capture the information about object features in their representations. Specifically, we train an MLP to predict the value of the $i$-th object feature given the frozen embeddings produced by a finetuned network initialized from (a) random initialization, (b) mask prediction pretraining, (c) target-aware mask prediction pretraining. The separate MLP is used for each feature and the RMSE learning objective is used. Then we report the RMSE on the test set for all features $i \in [0, m]$ along with their importance rank in the dataset on Figure 1 (we use CatBoost feature importances). Here, the lower ranks correspond to the more important features. Figure 1 reveals that the target-aware pretraining enables the model to capture more information about the informative features compared to the "unsupervised" pretraining and, especially, to the learning from scratch. The latter one successfully captures the most informative feature from the training data, while suppressing the less important, but still significant features. We conjecture that this is the source of superiority of the target-aware pretraining.

## 5.2 Efficient ensembling

Here we show, that it is possible to construct ensembles from one pretraining checkpoint (pretrained with the target conditioned mask prediction objective). To this end, we run finetuning with 15 different random seeds starting from the one pretrained checkpoint. Table 5 shows the results.

Both ensembling the models from a shared pretrain checkpoint and ensembling multiple independent pretraining runs produces strong ensembles, which shows that it is sufficient to pretrain once and create ensembles by several independent finetuning processes. This is important in practice since finetuninig is typically cheaper (i.e. requires fewer iterations), and still is able to produce diverse models from the one pretraining checkpoint for ensembles of comparable quality.

---

[3]Computed with the CatBoost method "`get_feature_importance()`"

|  | GE ↑ | CH ↑ | CA ↓ | HO ↓ | OT ↓ | HI ↑ | FB ↓ | AD ↑ | WE ↓ | CO ↑ | MI ↓ |
|---|---|---|---|---|---|---|---|---|---|---|---|
| no pretraining | 0.635 | 0.849 | 0.506 | 3.156 | 0.479 | 0.801 | 5.737 | 0.908 | 1.909 | 0.963 | 0.749 |
| aug | 0.693 | 0.856 | 0.441 | 3.077 | 0.459 | 0.814 | 5.689 | 0.914 | 1.883 | 0.968 | 0.748 |
| aug \| no finetune | 0.674 | 0.853 | 0.464 | 3.251 | 0.461 | 0.808 | 6.167 | 0.910 | 1.938 | 0.958 | 0.752 |
| rec + sup | 0.684 | 0.854 | 0.436 | 3.012 | 0.456 | 0.815 | 5.672 | 0.911 | 1.862 | 0.967 | 0.747 |
| rec + sup \| no finetune | 0.683 | 0.853 | 0.467 | 3.232 | 0.474 | 0.811 | 6.044 | 0.907 | 1.901 | 0.956 | 0.752 |

Table 6: Finetuning MLP on clean data versus using the model trained on corrupted inputs only.

|  | GE ↑ | CH ↑ | CA ↓ | HO ↓ | OT ↓ | HI ↑ | FB ↓ | AD ↑ | WE ↓ | CO ↑ | MI ↓ |
|---|---|---|---|---|---|---|---|---|---|---|---|
| CatBoost | 0.683 | 0.864 | 0.433 | 3.115 | 0.457 | 0.806 | 5.324 | 0.927 | 1.837 | 0.966 | 0.743 |
|  | 54s | 3s | 6s | 6s | 494s | 34s | 214s | 32s | 169s | 1869s | 1317s |
| MLP (no pretraining) | 0.635 | 0.849 | 0.506 | 3.156 | 0.479 | 0.801 | 5.737 | 0.908 | 1.909 | 0.963 | 0.749 |
|  | 29s | 10s | 25s | 26s | 38s | 29s | 159s | 12s | 57s | 352s | 211s |
| MLP (rec + sup \| 50k) | 0.679 | 0.857 | 0.441 | 3.064 | 0.462 | 0.813 | 5.650 | 0.910 | 1.879 | 0.966 | 0.747 |
|  | 327s | 227s | 280s | 277s | 355s | 256s | 624s | 403s | 242s | 593s | 292s |
| MLP (rec + sup \| 150k) | 0.692 | 0.859 | 0.435 | 3.012 | 0.456 | 0.816 | 5.629 | 0.910 | 1.866 | 0.968 | 0.746 |
|  | 891s | 338s | 758s | 509s | 712s | 862s | 916s | 1069s | 663s | 927s | 665s |

Table 7: Training times for CatBoost and MLP. Second row in each group reports average time spent training one model in seconds on an A100 GPU for MLPs and on a CPU for CatBoost.

### 5.3 ON IMPORTANCE OF FINETUNING ON CLEAN DATA

Here we show, that the second stage of finetuning the model on the entire dataset without input corruption is often necessary for the best downstream performance. To this end we compare finetuning the models on clean data with using models right after pretraining for two objectives: supervised loss and supervised loss with the reconstruction objective.

Results are shown in Table 6. Across all datasets for both methods finetuning on uncorrupted data with the supervised loss proves to be essential for the best performance. Sometimes excluding the second finetuning stage degrades the performance below the tuned supervised baseline of training from scratch.

### 5.4 DOES PRETRAINING REQUIRE MORE COMPUTE?

In this section we investigate how much more time is spent on pretraining, compared to training the models from scratch. We run pretraining with "rec + sup" objective with 50k and 150k pretraining iterations thresholds (early-stopping, in theory, could make 150k iterations equivalent to the 50k, but in practice it was not the case). We report downstream performance along with average time spent to pretrain and finetune a model in Table 7 (we also add CatBoost training times to the table).

Pretraining often requires by an order of magnitude more compute, it is especially apparent on smaller scale datasets like GE, CH, CA, HO, OT, HI, AD. However, the absolute time spent on pretraining is still acceptable, as the original training from scratch takes seconds on small datasets. Generally, the more iterations you use for pretraining, the better downstream quality you get.

## 6 CONCLUSION

In this work, we have systematically evaluated typical pretraining objectives for tabular deep learning. We have revealed several important recipes for optimal pretraining performance that can be universally beneficial across various problems and models. Our findings confirm that pretraining can significantly improve the performance of tabular deep models and provide additional evidence that tabular DL can become a strong alternative to GBDT.

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

## A   DATASETS

We used the following datasets:

- Gesture Phase Prediction (Madeo et al. (2013))
- Churn Modeling[4]
- California Housing (real estate data, Kelley Pace & Barry (1997))
- House 16H[5]
- Adult (income estimation, Kohavi (1996))
- Otto Group Product Classification[6]
- Higgs (simulated physical particles, Baldi et al. (2014); we use the version with 98K samples available at the OpenML repository Vanschoren et al. (2014))
- Facebook Comments (Singh et al. (2015))
- Covertype (forest characteristics, Blackard & Dean. (2000))
- Microsoft (search queries, Qin & Liu (2013)). We follow the pointwise approach to learning-to-rank and treat this ranking problem as a regression problem.
- Weather (temperature, Malinin et al. (2021)). We take 10% of the dataset for our experiments due to the its large size.

## B   HYPERPARAMETERS

### B.1   CATBOOST

We fix and do not tune the following hyperparameters:

- `early-stopping-rounds` $= 50$
- `od-pval` $= 0.001$
- `iterations` $= 2000$

For tuning on the MI and CO datasets, we set the `task_type` parameter to "GPU". In all other cases (including the evaluation on these two datasets), we set this parameter to "CPU".

---

[4]https://www.kaggle.com/shrutimechlearn/churn-modelling
[5]https://www.openml.org/d/574
[6]https://www.kaggle.com/c/otto-group-product-classification-challenge/data

| Parameter | Distribution |
|---|---|
| Max depth | $\mathrm{UniformInt}[1, 10]$ |
| Learning rate | $\mathrm{LogUniform}[0.001, 1]$ |
| Bagging temperature | $\mathrm{Uniform}[0, 1]$ |
| L2 leaf reg | $\mathrm{LogUniform}[1, 10]$ |
| Leaf estimation iterations | $\mathrm{UniformInt}[1, 10]$ |
| # Iterations | 100 |

Table 8: CatBoost hyperparameter space

## B.2 XGBOOST

We fix and do not tune the following hyperparameters:

- `booster` $=$ "gbtree"
- `early-stopping-rounds` $= 50$
- `n-estimators` $= 2000$

| Parameter | Distribution |
|---|---|
| Max depth | $\mathrm{UniformInt}[3, 10]$ |
| Min child weight | $\mathrm{LogUniform}[0.0001, 100]$ |
| Subsample | $\mathrm{Uniform}[0.5, 1]$ |
| Learning rate | $\mathrm{LogUniform}[0.001, 1]$ |
| Col sample by tree | $\mathrm{Uniform}[0.5, 1]$ |
| Gamma | $\{0, \mathrm{LogUniform}[0.001, 100]\}$ |
| Lambda | $\{0, \mathrm{LogUniform}[0.1, 10]\}$ |
| # Iterations | 100 |

Table 9: XGBoost hyperparameter space.

## B.3 MLP

We fix and do not tune the following hyperparameters:

- `Layer size` $= 512$
- `Head hidden size` $= 512$

| Parameter | Distribution |
|---|---|
| # Layers | $\mathrm{UniformInt}[1, 8]$ |
| Dropout | $\{0, \mathrm{Uniform}[0, 0.5]\}$ |
| Learning rate | $\mathrm{LogUniform}[5e\text{-}5, 0.005]$ |
| Weight decay | $\{0, \mathrm{LogUniform}[1e\text{-}6, 1e\text{-}3]\}$ |
| Corrupt Probability | $\mathrm{Uniform}[0.2, 0.8]$ |
| # Iterations | 100 |

Table 10: MLP hyperparameter space.

### B.4 EMBEDDING HYPERPARAMETERS

We fix and do not tune the following hyperparameters:

- Layer size $= 512$
- Head hidden size $= 512$

The distribution for the output dimensions of linear layers is $\text{UniformInt}[1, 128]$.

PLR, T-LR. We share the same hyperparameter space for models with embeddings across all datasets.

For the target-aware embeddings (tree-based) T-LR, the distribution for the number of leaves is $\text{UniformInt}[2, 256]$, the distribution for the minimum number of items per leaf is $\text{UniformInt}[1, 128]$ and the distribution for the minimum information gain required for making a split is $\text{LogUniform}[1e\text{-}9, 0.01]$.

For the periodic embeddings PLR. The distribution for $k$ is $\text{UniformInt}[1, 128]$, the distribution for the $\sigma$ parameter is $\text{LogUniform}[0.01, 100]$

## C SYNTHETIC DATASET DETAILS

To generate the corresponding objects targets $\{y_i\}_{i=1}^n$ for randomly sampled objects features $\{x_i\}_{i=1}^n$ and importance vector $p \in \mathbb{R}^m$ we build an ensemble of 10 random oblivious decision trees $\{T_i(x)\}_{i=1}^{10}$ of depth 10, where on each tree level we sample a feature $j \sim \text{Cat}(p)$ and a threshold $t \sim \text{Unif}(\min(x^j), \max(x^j))$ for a decision rule. For each tree leaf, we sample a scalar $l \sim \mathcal{N}(0, 1)$, representing a logit in binary classification. We define the binary targets as follows: $y(x) = \text{I}\left\{\frac{1}{10} \sum_{i=1}^{10} T_i(x) > 0\right\}$. We set the size of the dataset $n = 50.000$ and the number of features $m = 8$ and generate 50 datasets with different feature importance vectors $p$ for the analysis.

## D PRETRAINING THE FT-TRANSFORMER MODEL

Here we evaluate the recommended pretraining objectives for pretraining the FT-Transformer model from Gorishniy et al. (2021). We evaluate pretraining for the FT-Transformer on a subset of datasets for the reconstruction, mask prediction, and their supervised variations (mask+target and rec+sup). The results are in the Table 11.

| | GE ↑ | CH ↑ | CA ↓ | HO ↓ | OT ↓ | HI ↑ | AD ↑ | Avg. Rank |
|---|---|---|---|---|---|---|---|---|
| **MLP-PLR** | | | | | | | | |
| no pretraining | 0.668 | 0.858 | 0.469 | **3.008** | 0.483 | 0.809 | 0.926 | $2.7 \pm 1.2$ |
| mask | 0.685 | **0.863** | 0.434 | **3.007** | 0.477 | 0.818 | 0.927 | $2.1 \pm 1.1$ |
| rec | 0.667 | 0.852 | 0.439 | **3.031** | 0.472 | 0.808 | 0.926 | $2.1 \pm 1.1$ |
| rec + supervised | **0.709** | 0.858 | 0.433 | **3.059** | **0.465** | 0.807 | 0.927 | $1.7 \pm 1.2$ |
| mask + target | 0.694 | **0.862** | **0.425** | **3.023** | 0.474 | 0.821 | **0.929** | $1.9 \pm 1.1$ |
| **FT-Transformer** | | | | | | | | |
| no pretraining | 0.631 | **0.860** | 0.470 | 3.240 | 0.479 | 0.809 | 0.912 | $3.7 \pm 1.5$ |
| mask | 0.696 | **0.864** | **0.429** | 3.113 | **0.466** | **0.845** | 0.916 | $1.9 \pm 1.4$ |
| rec | 0.687 | **0.860** | 0.438 | 3.120 | 0.478 | 0.840 | 0.917 | $2.6 \pm 1.3$ |
| rec + supervised | **0.718** | **0.862** | **0.430** | 3.078 | **0.463** | 0.825 | 0.917 | $1.9 \pm 1.1$ |
| mask + target | 0.694 | **0.861** | 0.436 | 3.083 | 0.467 | **0.842** | 0.918 | $2.0 \pm 0.9$ |

Table 11: Results for pretraining the FT-Transformer model, along with the results for pretraining the MLP-PLR. Bold accounts for the best-performing model between both MLP-PLR and FT-Transformer.

We can see that pretraining provides performance improvements for the FT-Transformer model comparable to those for MLP-PLR, with MLP-PLR remaining state-of-the-art with or without pre-

training. Including transformers does not reveal new patterns or behavior, but provides additional evidence that pretraining does help neural networks on tabular data.

## E    COMPARISON TO THE LEARNING RATE SCHEDULERS

Since in our setup pretraining is a technique which helps in the optimization process, we compare it to various commonly used learning rate schedules which are also used as an aid in the optimization. The results for various learning rate schedulers along with two self-training based pretraining baselines are reported in Table 12.

| | GE↑ | CH↑ | CA↓ | HO↓ | OT↓ | HI↑ | FB↓ | AD↑ | WE↓ | CO↑ | MI↓ | Avg. Rank |
|---|---|---|---|---|---|---|---|---|---|---|---|---|
| no pretraining | 0.635 | 0.849 | 0.506 | 3.156 | 0.479 | 0.801 | 5.737 | 0.908 | 1.909 | 0.963 | 0.749 | 2.7 ± 0.6 |
| cyclic | 0.616 | 0.850 | 0.498 | 3.148 | 0.479 | 0.801 | 5.835 | 0.908 | 1.920 | 0.962 | 0.749 | 3.2 ± 0.9 |
| warmup | 0.655 | **0.854** | 0.491 | 3.123 | 0.478 | 0.799 | **5.662** | 0.908 | 1.912 | 0.964 | 0.749 | 2.5 ± 1.1 |
| warmup + lin | 0.669 | 0.851 | 0.489 | 3.123 | 0.480 | 0.801 | **5.669** | 0.908 | 1.912 | **0.968** | 0.748 | 2.2 ± 0.8 |
| warmup + cos | 0.639 | 0.851 | 0.494 | 3.107 | 0.478 | 0.801 | 5.823 | 0.908 | 1.893 | 0.962 | 0.748 | 2.6 ± 0.6 |
| mask | **0.691** | **0.857** | **0.454** | 3.113 | 0.472 | **0.814** | **5.681** | **0.912** | 1.883 | 0.964 | 0.748 | 1.5 ± 0.7 |
| rec | 0.662 | 0.853 | **0.445** | **3.044** | **0.466** | 0.805 | **5.641** | **0.910** | **1.875** | 0.965 | **0.746** | 1.5 ± 0.5 |

Table 12: Comparison of self-prediction based pretrains to learning rate schedulers. All the results are for the MLP model

We can see that the learning rate schedules provide some improvement and help the optimization process, however, the effect is almost negligible compared to the one from pretraining. We conjecture that exhaustive hyperparameter tuning diminishes the benefits of using the advanced learning rate schedules.

## F    DATASETS WITH MORE CATEGORICAL FEATURES

Additionally, we experiment with datasets that contain more categorical features. We add 4 datasets, the information about the datasets is presented in Table 13.

| Abbr | Name | # Train | # Validation | # Test | # Num | # Cat | Task type | Batch size |
|---|---|---|---|---|---|---|---|---|
| DI | Diamond | 34521 | 8631 | 10788 | 6 | 3 | Regression | 128 |
| BL | Black Friday | 106764 | 26692 | 33365 | 5 | 4 | Regression | 128 |
| BR | Brazilian houses | 6842 | 1710 | 2136 | 8 | 4 | Regression | 128 |
| BA | Bank | 28934 | 7234 | 9043 | 7 | 9 | Binclass | 128 |

Table 13: Datasets with categorical features

The results for the MLP and MLP-PLR with self-prediction-based pretrains along with their supervised variations are presented in Table 14.

We can see that results on the datasets with more categorical features align with the results in the main text: pretraining strategies based on self-prediction improve upon the supervised baselines and target-aware objectives help further increase the downstream performance.

## G    SHARE OR SPLIT LEARNING RATE AND WEIGHT DECAY BETWEEN PRETRAINING AND FINETUNING?

Here we demonstrate that tuning and using the same learning rate and weight decay for both pretraining and finetuning results in similar performance to tuning these parameters separately for the two stages. We opt for sharing the learning rate and weight decay for pretraining and finetuning in all the experiments in the paper.

|  | DI ↓ | BL ↓ | BR ↓ | BA ↑ |
|---|---|---|---|---|
| CatBoost | 0.134 | 0.682 | 0.407 | **0.932** |
| XGBoost | 0.136 | **0.680** | 0.407 | 0.930 |
| MLP | | | | |
| no pretraining | 0.140 | 0.697 | 0.406 | 0.927 |
| rec | 0.140 | 0.695 | 0.400 | 0.926 |
| mask | 0.138 | 0.691 | **0.385** | 0.929 |
| rec + supervised | 0.135 | 0.689 | 0.394 | 0.930 |
| mask + target | 0.135 | 0.691 | **0.376** | **0.933** |
| MLP-PLR | | | | |
| no pretraining | 0.134 | 0.689 | 0.399 | 0.929 |
| rec | 0.133 | 0.684 | 0.390 | 0.931 |
| mask | **0.131** | 0.685 | 0.392 | 0.931 |
| rec + supervised | 0.133 | 0.681 | 0.393 | 0.928 |
| mask + target | **0.132** | 0.685 | **0.385** | **0.933** |

Table 14: The results for single models with various pretraining objectives on datasets with categorical features.

|  | GE ↑ | CH ↑ | CA ↓ | HO ↓ | OT ↓ | HI ↑ | FB ↓ | AD ↑ | WE ↓ | CO ↑ | MI ↓ |
|---|---|---|---|---|---|---|---|---|---|---|---|
| shared wd / shared lr | $0.683_{\pm1e\text{-}2}$ | $0.857_{\pm2e\text{-}3}$ | $0.434_{\pm7e\text{-}3}$ | $3.056_{\pm4e\text{-}2}$ | $0.468_{\pm2e\text{-}3}$ | $0.819_{\pm2e\text{-}3}$ | $5.633_{\pm4e\text{-}2}$ | $0.914_{\pm1e\text{-}3}$ | $1.876_{\pm5e\text{-}3}$ | $0.965_{\pm7e\text{-}4}$ | $0.748_{\pm4e\text{-}4}$ |
| shared wd / split lr | $0.697_{\pm9e\text{-}3}$ | $0.857_{\pm3e\text{-}3}$ | $0.431_{\pm7e\text{-}3}$ | $3.032_{\pm3e\text{-}2}$ | $0.469_{\pm2e\text{-}3}$ | $0.819_{\pm2e\text{-}3}$ | $5.647_{\pm4e\text{-}2}$ | $0.915_{\pm9e\text{-}4}$ | $1.934_{\pm8e\text{-}3}$ | $0.964_{\pm9e\text{-}4}$ | $0.748_{\pm4e\text{-}4}$ |
| split wd / split lr | $0.688_{\pm9e\text{-}3}$ | $0.856_{\pm3e\text{-}3}$ | $0.430_{\pm4e\text{-}3}$ | $3.046_{\pm4e\text{-}2}$ | $0.471_{\pm3e\text{-}3}$ | $0.821_{\pm7e\text{-}4}$ | $5.734_{\pm5e\text{-}2}$ | $0.914_{\pm7e\text{-}4}$ | $1.891_{\pm6e\text{-}3}$ | $0.964_{\pm1e\text{-}3}$ | $0.748_{\pm3e\text{-}4}$ |
| split wd / split lr | $0.694_{\pm1e\text{-}2}$ | $0.858_{\pm2e\text{-}3}$ | $0.431_{\pm6e\text{-}3}$ | $3.066_{\pm3e\text{-}2}$ | $0.468_{\pm2e\text{-}3}$ | $0.821_{\pm2e\text{-}3}$ | $5.632_{\pm4e\text{-}2}$ | $0.914_{\pm1e\text{-}3}$ | $1.878_{\pm4e\text{-}3}$ | $0.966_{\pm1e\text{-}3}$ | $0.748_{\pm3e\text{-}4}$ |

Table 15: Results for single models with MLP mask + target pretraining

## H   EARLY-STOPPING CRITERIONS

Here we demonstrate that early stopping the pretraining by the value of the pretraining objective on the hold-out validation set is comparable to the early stopping by the downstream metric on the hold-out validation set after finetuning. See Table 16 for the results. This is an important practical observation, as computing pretraining objective is much faster than the full finetuning of the model, especially on large scale datasets.

|  | GE ↑ | CH ↑ | CA ↓ | HO ↓ | OT ↓ | HI ↑ | FB ↓ | AD ↑ |
|---|---|---|---|---|---|---|---|---|
| finetune early stop | 0.683 | 0.857 | 0.434 | 3.056 | 0.468 | 0.819 | 5.633 | 0.914 |
| pretrain early stop | 0.674 | 0.855 | 0.434 | 3.031 | 0.469 | 0.818 | 5.738 | 0.914 |

Table 16: Results for single models with MLP mask + target pretraining

## I   APPLYING AUGMENTATIONS TO GBDTS

We adopt the augmentations used in pretraining to GBDTs by augmenting the datasets before GBDT tree construction, effectively enlarging datasets with random column resampling to match the number of perturbations the neural network is exposed to during pretraining (tens of millions of randomly corrupted samples). We report results in Table 17. We observe that adding noisy augmented samples to the datasets deteriorates GBDT performance significantly.

## J   SUMMARY OF PRETRAINING METHODS

For summary of the pretraining methods mentioned in the paper see Table 18.

|  | GE ↑ | CH ↑ | CA ↓ | HO ↓ | AD ↑ | HI ↑ |
|---|---|---|---|---|---|---|
| CatBoost | 0.682 | 0.864 | 0.433 | 3.115 | 0.927 | 0.806 |
| CatBoost (aug) | 0.660 | 0.838 | 0.474 | 3.561 | 0.923 | 0.791 |

Table 17: Comparison of CatBoost and CatBoost on data augmented with random column resampling.

| Abbr. | Description | Uses target variable |
|---|---|---|
| contrastive | Contrastive loss with resampling as augmentation | ✗ |
| mask | Prediction of masks for the resampled columns | ✗ |
| rec | Reconstruction original values of resampled columns | ✗ |
| mask + sup | Mask prediction combined with supervised loss | ✓ |
| mask + target | Mask prediction with modified column resampling and loss | ✓ |
| rec + sup | Reconstruction combined with supervised loss | ✓ |
| rec + target | Reconstruction with modified column resampling and loss | ✓ |

Table 18: Summary of the objectives studied in this work

# K    EXTENDED TABLES WITH EXPERIMENTAL RESULTS

The scores with standard deviations for single models and ensembles are provided in 19 and 20 respectively.

| | GE↑ | CH↑ | CA↓ | HO↓ | OT↓ | HI↑ | FB↓ | AD↑ | WE↓ | CO↑ | MI↓ |
|---|---|---|---|---|---|---|---|---|---|---|---|
| CatBoost | 0.683±4.7e-3 | 0.864±8.1e-4 | 0.433±1.7e-3 | 3.115±1.8e-2 | 0.457±1.3e-3 | 0.806±3.4e-3 | 5.324±4.0e-2 | 0.927±3.1e-4 | 1.837±2.1e-4 | 0.966±3.2e-4 | 0.743±3.0e-4 |
| XGBoost | 0.678±4.8e-3 | 0.858±2.3e-3 | 0.436±2.5e-3 | 3.160±6.9e-3 | 0.457±6.0e-3 | 0.804±1.5e-3 | 5.383±2.8e-2 | 0.927±7.0e-4 | 1.802±2.0e-3 | 0.969±6.1e-4 | 0.742±1.5e-4 |
| **MLP** | | | | | | | | | | | |
| no pretraining | 0.635±1.3e-2 | 0.849±1.6e-3 | 0.506±8.6e-3 | 3.156±2.1e-2 | 0.479±1.4e-2 | 0.801±9.5e-4 | 5.737±6.1e-2 | 0.908±1.0e-3 | 1.909±4.6e-3 | 0.963±8.7e-4 | 0.749±3.6e-4 |
| mask | 0.691±1.0e-2 | 0.857±2.5e-3 | 0.454±5.0e-3 | 3.113±4.3e-2 | 0.472±3.0e-2 | 0.814±1.7e-3 | 5.681±3.1e-2 | 0.912±8.0e-4 | 1.883±2.9e-3 | 0.964±9.3e-4 | 0.748±3.1e-4 |
| rec | 0.662±1.0e-2 | 0.853±2.2e-3 | 0.445±4.0e-3 | 3.044±3.3e-2 | 0.466±2.1e-3 | 0.805±1.3e-3 | 5.641±3.2e-2 | 0.910±1.2e-3 | 1.875±3.4e-3 | 0.965±5.4e-4 | 0.746±2.3e-4 |
| contrastive | 0.672±1.4e-2 | 0.855±2.0e-3 | 0.455±4.5e-3 | 3.056±5.5e-2 | 0.469±2.6e-3 | 0.813±1.3e-3 | 5.697±2.9e-2 | 0.910±1.3e-3 | 1.881±4.5e-3 | 0.960±1.2e-3 | 0.748±3.6e-4 |
| sup | 0.693±1.1e-2 | 0.856±1.8e-3 | 0.441±5.3e-3 | 3.077±2.9e-2 | 0.459±2.1e-3 | 0.814±7.4e-4 | 5.689±2.1e-2 | 0.914±7.8e-4 | 1.883±5.2e-3 | 0.968±5.2e-4 | 0.748±3.0e-4 |
| supcon | 0.666±1.4e-2 | 0.850±2.2e-3 | 0.454±4.2e-3 | 3.108±2.5e-2 | 0.480±1.9e-3 | 0.806±7.3e-4 | 5.680±2.4e-2 | 0.911±6.0e-4 | 1.873±2.4e-3 | 0.966±5.8e-4 | 0.747±3.2e-4 |
| mask + sup | 0.693±8.2e-3 | 0.857±2.3e-3 | 0.436±6.9e-3 | 3.099±2.4e-2 | 0.458±1.7e-3 | 0.817±5.6e-4 | 5.685±3.6e-2 | 0.915±5.3e-4 | 1.873±5.1e-3 | 0.967±4.0e-4 | 0.748±2.8e-4 |
| rec + sup | 0.684±7.7e-3 | 0.854±4.5e-3 | 0.436±4.4e-3 | 3.012±4.0e-2 | 0.456±1.9e-3 | 0.815±5.8e-4 | 5.672±3.6e-2 | 0.911±1.4e-3 | 1.862±2.8e-3 | 0.967±6.6e-4 | 0.747±4.9e-4 |
| mask + target | 0.683±1.0e-2 | 0.857±2.1e-3 | 0.434±7.2e-3 | 3.056±4.0e-2 | 0.468±1.9e-3 | 0.819±1.6e-3 | 5.633±3.7e-2 | 0.914±1.1e-3 | 1.876±4.8e-3 | 0.965±6.6e-4 | 0.748±4.5e-4 |
| - target sampling | 0.680±9.7e-3 | 0.857±3.1e-3 | 0.432±4.9e-3 | 3.019±3.5e-2 | 0.468±1.9e-3 | 0.815±1.7e-3 | 5.697±3.1e-2 | 0.912±6.7e-4 | 1.887±3.1e-3 | 0.964±1.1e-3 | 0.748±3.1e-4 |
| rec + target | 0.659±8.6e-3 | 0.853±3.2e-3 | 0.454±6.7e-3 | 3.044±4.9e-2 | 0.463±1.6e-3 | 0.806±1.5e-3 | 5.636±3.1e-2 | 0.909±9.0e-4 | 1.884±2.3e-3 | 0.965±8.7e-4 | 0.745±3.9e-4 |
| - target sampling | 0.641±5.6e-3 | 0.853±3.4e-3 | 0.455±4.6e-3 | 3.046±2.4e-2 | 0.463±2.4e-2 | 0.806±1.3e-3 | 5.640±1.9e-2 | 0.910±1.1e-3 | 1.877±3.3e-3 | 0.966±4.4e-4 | 0.746±3.9e-4 |
| **MLP-PLR** | | | | | | | | | | | |
| no pretraining | 0.668±1.4e-2 | 0.858±4.7e-3 | 0.469±5.2e-3 | 3.008±2.3e-2 | 0.483±1.6e-3 | 0.809±2.3e-3 | 5.608±5.6e-2 | 0.926±6.1e-4 | 1.890±5.0e-3 | 0.969±1.0e-3 | 0.746±3.7e-4 |
| mask | 0.685±5.6e-3 | 0.863±1.8e-3 | 0.434±4.3e-3 | 3.007±4.6e-2 | 0.477±2.5e-3 | 0.818±8.4e-4 | 5.586±2.4e-2 | 0.927±5.1e-4 | 1.911±5.4e-4 | 0.970±5.1e-4 | 0.748±3.9e-4 |
| rec | 0.667±7.2e-3 | 0.852±7.2e-3 | 0.439±5.2e-3 | 3.031±3.8e-2 | 0.472±3.0e-3 | 0.808±1.2e-3 | 5.571±1.2e-1 | 0.926±6.4e-4 | 1.877±4.0e-3 | 0.971±4.8e-4 | 0.745±3.6e-4 |
| sup | 0.710±4.1e-3 | 0.859±4.1e-3 | 0.433±3.6e-3 | 3.136±8.1e-2 | 0.479±1.9e-3 | 0.811±1.0e-3 | 5.521±4.6e-2 | 0.924±1.5e-3 | 1.873±2.0e-3 | 0.971±1.0e-3 | 0.748±8.0e-4 |
| mask + sup | 0.711±7.1e-3 | 0.866±2.0e-3 | 0.441±3.6e-3 | 3.129±4.1e-2 | 0.480±1.8e-3 | 0.813±8.2e-4 | 5.480±4.6e-2 | 0.925±1.0e-3 | 1.875±2.2e-3 | 0.969±6.0e-4 | 0.745±2.4e-4 |
| rec + sup | 0.709±5.1e-3 | 0.858±1.9e-3 | 0.433±2.7e-3 | 3.059±3.6e-2 | 0.465±2.2e-3 | 0.807±6.2e-4 | 5.571±1.2e-1 | 0.927±5.8e-4 | 1.865±3.1e-3 | 0.971±4.4e-4 | 0.745±2.4e-4 |
| mask + target | 0.694±9.1e-3 | 0.862±1.7e-3 | 0.425±4.2e-3 | 3.023±4.3e-2 | 0.474±2.0e-3 | 0.821±1.1e-3 | 5.537±3.4e-2 | 0.929±3.3e-4 | 1.911±6.2e-3 | 0.969±5.8e-4 | 0.749±1.2e-3 |
| - target sampling | 0.690±9.6e-3 | 0.864±2.8e-3 | 0.421±4.5e-3 | 2.971±4.1e-2 | 0.479±2.0e-3 | 0.821±1.0e-3 | 5.440±8.1e-2 | 0.928±5.6e-4 | 1.906±5.5e-3 | 0.970±3.9e-4 | 0.748±5.3e-4 |
| rec + target | 0.688±8.2e-3 | 0.860±1.4e-3 | 0.445±2.7e-3 | 3.064±3.4e-2 | 0.475±1.9e-3 | 0.812±1.1e-3 | 5.507±1.0e-1 | 0.927±3.9e-4 | 1.887±2.4e-3 | 0.971±3.9e-4 | 0.748±7.2e-4 |
| - target sampling | 0.687±7.9e-3 | 0.855±4.8e-3 | 0.453±6.4e-3 | 3.008±2.7e-2 | 0.471±2.1e-3 | 0.812±5.5e-4 | 5.592±9.8e-2 | 0.927±3.4e-4 | 1.876±3.1e-3 | 0.970±4.4e-4 | 0.745±3.5e-4 |
| **MLP-T-LR** | | | | | | | | | | | |
| no pretraining | 0.634±6.9e-3 | 0.866±1.3e-3 | 0.444±1.8e-3 | 3.113±4.5e-2 | 0.482±1.7e-3 | 0.805±9.3e-4 | 5.520±3.6e-2 | 0.925±6.8e-4 | 1.897±4.5e-3 | 0.968±5.0e-4 | 0.749±5.2e-4 |
| mask | 0.654±6.4e-3 | 0.868±1.0e-3 | 0.424±2.4e-3 | 3.045±3.7e-2 | 0.472±2.5e-3 | 0.818±1.8e-3 | 5.544±3.5e-2 | 0.926±7.1e-4 | 1.916±3.1e-3 | 0.969±4.6e-4 | 0.748±3.5e-4 |
| rec | 0.652±7.4e-3 | 0.857±4.4e-3 | 0.424±3.1e-3 | 3.109±3.7e-2 | 0.472±2.0e-3 | 0.808±1.0e-3 | 5.363±6.6e-2 | 0.924±1.8e-4 | 1.861±3.9e-3 | 0.969±7.3e-4 | 0.746±4.8e-4 |
| sup | 0.682±5.1e-3 | 0.860±4.1e-3 | 0.430±1.9e-3 | 3.135±1.7e-2 | 0.471±1.2e-3 | 0.807±6.2e-4 | 5.525±2.3e-2 | 0.927±3.8e-4 | 1.893±2.4e-3 | 0.971±5.8e-4 | 0.747±3.1e-4 |
| mask + sup | 0.676±7.7e-3 | 0.858±7.9e-3 | 0.429±1.8e-3 | 3.199±2.4e-2 | 0.468±9.9e-4 | 0.814±8.2e-4 | 5.510±3.6e-2 | 0.926±8.7e-4 | 1.869±2.9e-3 | 0.971±3.9e-4 | 0.748±2.5e-4 |
| rec + sup | 0.678±6.7e-3 | 0.865±1.1e-3 | 0.437±2.0e-3 | 3.112±4.1e-2 | 0.462±2.2e-3 | 0.807±2.6e-3 | 5.516±2.2e-2 | 0.927±3.7e-4 | 1.862±3.4e-3 | 0.970±5.0e-4 | 0.748±4.2e-4 |
| mask + target | 0.649±7.3e-3 | 0.865±1.7e-3 | 0.421±3.9e-3 | 3.058±4.3e-2 | 0.474±1.9e-3 | 0.820±1.2e-3 | 5.644±4.6e-2 | 0.929±3.5e-4 | 1.924±7.6e-3 | 0.969±4.8e-4 | 0.749±5.1e-4 |
| - target sampling | 0.649±8.3e-3 | 0.861±3.2e-3 | 0.417±4.9e-3 | 3.050±3.3e-2 | 0.476±1.7e-3 | 0.819±1.7e-3 | 5.492±3.2e-2 | 0.928±5.6e-4 | 1.874±3.7e-3 | 0.969±4.0e-4 | 0.749±1.4e-3 |
| rec + target | 0.668±7.0e-3 | 0.864±1.7e-3 | 0.440±3.5e-3 | 3.113±3.5e-2 | 0.473±4.3e-3 | 0.806±1.0e-3 | 5.493±4.4e-2 | 0.927±5.6e-4 | 1.862±3.5e-3 | 0.969±6.5e-4 | 0.746±3.4e-4 |
| - target sampling | 0.667±1.0e-2 | 0.859±2.7e-3 | 0.432±3.3e-3 | 3.104±2.9e-2 | 0.470±2.2e-3 | 0.806±1.5e-3 | 5.391±3.9e-2 | 0.927±4.1e-4 | 1.867±3.8e-3 | 0.968±7.5e-4 | 0.746±3.1e-4 |

Table 19: Extended results for single models

| | GE↑ | CH↑ | CA↓ | HO↓ | OT↓ | HI↑ | FB↓ | AD↑ | WE↓ | CO↑ | MI↓ |
|---|---|---|---|---|---|---|---|---|---|---|---|
| CatBoost | 0.692±1.8e-3 | 0.864±6.8e-5 | 0.430±1.1e-3 | 3.093±5.1e-3 | 0.450±3.5e-4 | 0.807±7.5e-5 | 5.226±1.2e-2 | 0.928±1.3e-4 | 1.801±1.2e-4 | 0.967±1.3e-4 | 0.741±1.4e-4 |
| XGBoost | 0.683±1.3e-3 | 0.860±4.3e-4 | 0.434±7.0e-4 | 3.152±1.2e-4 | 0.454±2.5e-4 | 0.805±8.3e-4 | 5.338±1.8e-2 | 0.927±2.1e-4 | 1.782±4.9e-4 | 0.969±8.8e-5 | 0.742±5.3e-5 |
| | | | | **MLP** | | | | | | | |
| no pretraining | 0.656±5.9e-3 | 0.852±5.2e-4 | 0.482±2.9e-4 | 3.055±8.4e-3 | 0.467±2.0e-3 | 0.805±2.9e-4 | 5.666±2.6e-3 | 0.910±2.7e-4 | 1.850±1.0e-3 | 0.968±2.5e-4 | 0.747±8.6e-5 |
| mask | 0.722±1.6e-3 | 0.859±6.4e-4 | 0.437±8.1e-4 | 3.026±6.3e-3 | 0.451±1.6e-3 | 0.824±6.5e-4 | 5.578±6.8e-3 | 0.913±3.0e-4 | 1.828±1.0e-3 | 0.967±1.1e-4 | 0.746±7.7e-5 |
| rec | 0.679±2.0e-3 | 0.856±2.8e-4 | 0.424±1.0e-4 | 2.967±6.9e-3 | 0.453±1.2e-3 | 0.812±2.5e-4 | 5.571±1.4e-2 | 0.912±7.5e-5 | 1.811±1.4e-3 | 0.972±2.0e-4 | 0.744±7.8e-5 |
| contrastive | 0.708±4.4e-3 | 0.857±8.8e-4 | 0.434±4.0e-3 | 2.952±4.4e-3 | 0.451±6.1e-4 | 0.820±4.5e-5 | 5.634±1.4e-2 | 0.912±1.3e-4 | 1.804±2.8e-3 | 0.964±2.1e-4 | 0.746±1.7e-4 |
| sup | 0.717±2.2e-3 | 0.857±7.4e-4 | 0.424±9.4e-4 | 3.022±1.9e-2 | 0.443±1.9e-3 | 0.816±1.4e-4 | 5.602±6.5e-3 | 0.916±7.0e-5 | 1.828±4.7e-3 | 0.973±3.6e-4 | 0.746±2.1e-4 |
| supcon | 0.686±5.2e-3 | 0.851±8.4e-4 | 0.434±3.0e-3 | 3.014±6.0e-3 | 0.465±1.3e-3 | 0.809±1.3e-3 | 5.579±3.4e-3 | 0.912±3.2e-4 | 1.827±9.4e-4 | 0.970±2.1e-4 | 0.745±5.3e-5 |
| mask + sup | 0.716±5.7e-3 | 0.859±1.1e-3 | 0.418±2.2e-3 | 3.066±1.2e-2 | 0.443±1.5e-3 | 0.819±1.3e-4 | 5.601±4.7e-3 | 0.916±2.0e-4 | 1.810±2.9e-4 | 0.973±3.9e-5 | 0.747±1.3e-4 |
| rec + sup | 0.709±3.7e-3 | 0.859±1.8e-3 | 0.419±2.1e-3 | 2.951±1.9e-2 | 0.442±8.6e-4 | 0.817±1.0e-4 | 5.531±3.0e-3 | 0.913±5.2e-4 | 1.801±4.0e-3 | 0.973±2.6e-5 | 0.745±1.1e-4 |
| mask + target | 0.709±7.3e-3 | 0.860±1.6e-4 | 0.414±1.1e-3 | 2.949±1.9e-2 | 0.457±5.9e-4 | 0.828±6.3e-4 | 5.551±7.2e-3 | 0.916±4.6e-4 | 1.809±5.3e-4 | 0.969±5.2e-5 | 0.746±1.9e-4 |
| - target sampling | 0.706±4.9e-3 | 0.860±1.1e-3 | 0.410±1.5e-3 | 2.955±2.1e-2 | 0.456±3.8e-4 | 0.822±1.1e-3 | 5.601±2.0e-2 | 0.914±2.5e-4 | 1.837±1.0e-3 | 0.968±2.8e-4 | 0.746±2.1e-4 |
| rec + target | 0.677±4.6e-3 | 0.857±3.9e-4 | 0.433±1.1e-3 | 2.926±2.3e-2 | 0.448±9.4e-4 | 0.816±5.3e-4 | 5.555±5.8e-3 | 0.910±2.2e-4 | 1.825±2.5e-3 | 0.972±6.1e-5 | 0.743±1.2e-4 |
| - target sampling | 0.669±3.7e-3 | 0.858±1.4e-4 | 0.435±8.2e-4 | 3.003±1.1e-2 | 0.451±3.4e-4 | 0.815±7.0e-4 | 5.577±1.1e-2 | 0.913±2.4e-4 | 1.822±6.6e-4 | 0.972±2.7e-4 | 0.744±7.6e-5 |
| | | | | **MLP-PLR** | | | | | | | |
| no pretraining | 0.695±3.7e-3 | 0.864±7.6e-4 | 0.454±1.2e-3 | 2.953±7.4e-3 | 0.470±7.5e-4 | 0.814±7.9e-4 | 5.324±3.2e-2 | 0.928±7.7e-5 | 1.835±1.5e-3 | 0.974±2.2e-4 | 0.744±1.2e-4 |
| mask | 0.725±4.9e-3 | 0.865±7.0e-4 | 0.421±1.7e-3 | 2.921±1.0e-2 | 0.457±8.4e-4 | 0.827±1.1e-4 | 5.444±5.4e-3 | 0.928±1.0e-4 | 1.850±3.4e-3 | 0.974±2.3e-4 | 0.745±6.5e-5 |
| rec | 0.698±1.5e-3 | 0.857±1.5e-3 | 0.418±1.2e-3 | 2.954±8.1e-3 | 0.454±1.9e-3 | 0.813±7.8e-4 | 5.124±2.4e-2 | 0.928±2.2e-4 | 1.818±2.5e-3 | 0.975±2.9e-4 | 0.743±2.1e-4 |
| sup | 0.733±2.2e-3 | 0.867±9.6e-4 | 0.421±1.2e-3 | 3.054±4.5e-2 | 0.465±1.3e-3 | 0.816±1.4e-4 | 5.407±3.8e-2 | 0.926±4.3e-4 | 1.834±3.3e-4 | 0.975±1.6e-4 | 0.746±2.1e-4 |
| mask + sup | 0.732±2.0e-3 | 0.869±3.5e-4 | 0.424±8.9e-4 | 3.055±1.6e-2 | 0.468±7.8e-4 | 0.817±3.4e-4 | 5.366±1.1e-2 | 0.927±2.1e-4 | 1.848±1.7e-3 | 0.974±7.9e-5 | 0.744±7.9e-5 |
| rec + sup | 0.737±2.0e-3 | 0.862±1.3e-3 | 0.424±9.9e-4 | 2.964±2.3e-3 | 0.449±2.3e-4 | 0.811±5.3e-4 | 5.124±2.4e-2 | 0.929±1.7e-4 | 1.813±1.9e-3 | 0.974±2.2e-4 | 0.744±7.2e-5 |
| mask + target | 0.719±3.5e-3 | 0.866±4.3e-4 | 0.407±8.2e-4 | 2.952±3.5e-3 | 0.458±4.2e-4 | 0.828±6.0e-4 | 5.373±1.8e-2 | 0.930±1.0e-4 | 1.849±2.1e-3 | 0.973±1.4e-4 | 0.745±2.5e-4 |
| - target sampling | 0.724±7.0e-3 | 0.867±1.0e-3 | 0.403±1.1e-3 | 2.877±1.0e-2 | 0.466±9.4e-4 | 0.828±2.5e-4 | 5.175±1.5e-2 | 0.930±2.0e-4 | 1.833±3.3e-4 | 0.974±3.3e-4 | 0.744±2.0e-4 |
| rec + target | 0.705±2.3e-3 | 0.862±2.0e-4 | 0.431±1.4e-3 | 2.983±1.2e-2 | 0.465±9.2e-4 | 0.816±1.7e-4 | 5.096±1.8e-2 | 0.928±1.9e-4 | 1.860±2.0e-3 | 0.974±3.4e-5 | 0.745±2.7e-4 |
| - target sampling | 0.712±2.3e-3 | 0.860±9.3e-4 | 0.437±3.3e-3 | 2.933±2.0e-2 | 0.450±1.7e-3 | 0.815±3.5e-4 | 5.173±2.7e-2 | 0.928±5.6e-5 | 1.811±1.5e-3 | 0.974±3.6e-4 | 0.744±6.3e-5 |
| | | | | **MLP-T-LR** | | | | | | | |
| no pretraining | 0.662±7.6e-3 | 0.868±5.0e-4 | 0.437±8.2e-4 | 3.028±1.8e-2 | 0.472±4.7e-4 | 0.808±1.5e-4 | 5.424±2.2e-2 | 0.927±2.8e-4 | 1.850±9.0e-4 | 0.972±1.5e-4 | 0.747±7.6e-5 |
| mask | 0.679±4.7e-3 | 0.868±2.3e-4 | 0.413±1.0e-3 | 2.930±1.2e-2 | 0.450±7.3e-4 | 0.826±1.3e-3 | 5.370±8.7e-3 | 0.927±3.1e-4 | 1.836±2.7e-3 | 0.973±8.8e-5 | 0.745±1.1e-4 |
| rec | 0.694±3.7e-3 | 0.861±1.6e-4 | 0.414±1.5e-3 | 3.035±1.9e-2 | 0.459±3.4e-4 | 0.812±2.7e-4 | 5.039±1.8e-2 | 0.925±1.2e-4 | 1.803±2.3e-3 | 0.973±4.9e-5 | 0.744±4.5e-4 |
| sup | 0.698±3.7e-3 | 0.865±7.0e-4 | 0.424±1.1e-3 | 3.107±6.8e-3 | 0.463±3.5e-4 | 0.809±2.5e-4 | 5.442±1.6e-2 | 0.928±1.4e-4 | 1.849±5.4e-4 | 0.975±2.8e-4 | 0.746±1.4e-5 |
| mask + sup | 0.698±4.0e-3 | 0.866±1.1e-3 | 0.421±8.6e-4 | 3.088±5.1e-3 | 0.460±4.6e-4 | 0.818±2.8e-4 | 5.407±3.3e-3 | 0.927±5.1e-4 | 1.824±3.0e-3 | 0.975±1.4e-4 | 0.747±8.2e-5 |
| rec + sup | 0.705±2.4e-3 | 0.866±4.6e-4 | 0.425±5.1e-4 | 3.057±1.0e-2 | 0.444±1.4e-3 | 0.814±6.8e-4 | 5.422±1.8e-3 | 0.927±6.6e-5 | 1.811±1.0e-3 | 0.974±1.1e-4 | 0.746±3.6e-4 |
| mask + target | 0.673±1.0e-3 | 0.868±4.6e-4 | 0.410±7.8e-4 | 2.894±1.8e-2 | 0.460±1.4e-3 | 0.827±3.4e-4 | 5.458±2.8e-2 | 0.930±1.7e-5 | 1.849±4.2e-3 | 0.972±2.5e-4 | 0.746±2.3e-4 |
| - target sampling | 0.677±5.1e-3 | 0.866±1.1e-3 | 0.397±3.7e-4 | 2.938±1.8e-2 | 0.462±4.4e-4 | 0.826±1.2e-4 | 5.384±1.5e-2 | 0.929±1.6e-4 | 1.840±2.2e-3 | 0.973±1.1e-4 | 0.747±5.0e-4 |
| rec + target | 0.693±2.5e-3 | 0.866±3.1e-4 | 0.432±6.1e-4 | 3.045±1.1e-2 | 0.456±2.4e-3 | 0.812±6.1e-4 | 5.344±8.1e-3 | 0.928±1.4e-4 | 1.830±2.4e-3 | 0.972±1.1e-4 | 0.744±2.1e-4 |
| - target sampling | 0.700±2.5e-3 | 0.863±1.3e-3 | 0.423±9.8e-4 | 3.029±1.4e-2 | 0.454±1.8e-3 | 0.811±5.1e-4 | 5.083±4.7e-3 | 0.928±1.7e-4 | 1.809±1.0e-3 | 0.972±5.5e-5 | 0.744±3.0e-5 |

Table 20: Extended results for ensemble models

