# OpenReview forum: "Revisiting Pretraining Objectives for Tabular Deep Learning"
_ICLR.cc/2023/Conference — Submitted to ICLR 2023_

### Official Review · Reviewer_MXDS · 2022-10-17

**Confidence:** 3
**Correctness:** 3
**Technical Novelty And Significance:** 2
**Empirical Novelty And Significance:** 3
**Recommendation:** 6

**Clarity, Quality, Novelty And Reproducibility:**

The clarity of the paper can be improved, but the paper provides a nice overview and evaluation adapting existing pretraining objectives from vision and NLP tasks to tabular tasks.

**Strength And Weaknesses:**

Strengths
---
Pretraining is an important part of the significant success of DL models on downstream vision and NLP tasks, and thus exploring various approaches of pretraining for tabular data is an important and potentially significant endeavor.

The insight of self-prediction generally outperforming contrastive learning is interesting and significant.

Experiments are performed on a wide range of tabular classification and regression datasets.

Weaknesses
---
There should be a more detailed comparison to GBDTs. The performance of the best pretraining methods are often similar or marginally better than the GBDT methods. An empirical runtime comparison of the training and inference times between GBDTs and the MLPs would provide significant value. Runtime results in addition to model size comparisons would give readers and machine learning practitioners a better sense of the tradeoffs between these methods and ultimately decide if the marginal increase in predictive performance is worth the additional complexity of different pretraining methods.

No comparisons to AutoML methods such as AutoGluon, which often outperform simple GBDT methods for tabular tasks.

The writing could be more precise in places. For example, what percentage of the inputs are corrupted in the description of the contrastive approach described in Section 3.2? Also, in Tables 3 and 4, the caption describes the bolded results as statistically significant across all models; however, there are multiple bolded numbers in the same column, thus I'm not sure what the bolded results are significant to. There are also a number of grammatical errors; consider using a service like Grammarly to fix these issues.

I think a more detailed discussion of the differences between this work and the work by Gorishniy et al. (2021) would greatly benefit readers. Also a table summarizing all of the pretraining methods discussed in this paper would also help readers significantly.

Minor Weaknesses
---
The definitions of "standard ensemble" and "efficient ensemble" in Table 5 are not clearly stated. There are also no standard errors or significance tests for Table 5.

Consider using "\cite" when passively citing previous work.

When open quotes are used, they are facing the wrong direction.

Tables 1, 6, and 7 are shown in the middle of a paragraph, consider moving them to either before or after the paragraph, or to the top or bottom of the page.

Missing period in the captions of Tables 1, 2, and 5.

"MLP-PLR" should be "MLP-P-LR" in the "Models" paragraph of Section 3.1


**Summary Of The Paper:**

The authors explore different pretraining objectives for tabular datasets, apply them to vanilla and embedding-based MLP architectures, evaluate their relative downstream predictive performance, and compare their performance with gradient-boosted decision trees (GBDTs, namely CatBoost and XGBoost). Specifically, the authors explore unsupervised pretraining objectives, target-aware pretraining objectives, and combinations of both.

Experiments on 11 datasets (6 classification and 5 regression tasks) suggest several findings: 1) deep learning (DL) models generally benefit from pretraining in terms of downstream performance (this includes both simple MLP and transformer-based architectures, although there was no improvement of transformer-based models over simple MLP architectures); 2) target-aware pretraining objectives performed better than unsupervised objectives; 3) combining target-aware and unsupervised pretraining objectives tends to work best, and perform similarly or better than GBDTs on downstream tasks.

**Summary Of The Review:**

The paper builds on work by Gorishniy et al. (2021) and expands the evaluation of pretraining objectives for tabular data. However, the paper lacks a more thorough comparison to GBDTs and AutoML methods, and the clarity of the paper could be improved.

---

> ### Author Response · Authors · 2022-11-14
> **Response to Reviewer MXDS**
>
> Thanks for the detailed review and interesting suggestions!
>
> **More comparison to GBDT**
>
> We agree that adding more comparisons to GBDT is important, we add an experiment, answering reviewer `QuBL` question about augmentations applied to gradient boosted decision trees.
>
> We also added training times for the GBDTs to the Table 7, in addition to pretraining times. Although, it's hard to compare training and inference times of GBDTs and Neural Networks, and control for all the factors:
> - Codebases are different (Pytorch/CatBoost)
> - CatBoost produces notably worse results, when accelerated on GPUs, for that reason we run it on CPUs
>
> Regardless, the reader could get the general idea from Table 7 – the absolute time spent training models is low, and it is comparable for both NNs and GBDTs. We also measure the inference speed for trained models on the CA dataset in the same setup (MLP run on a GPU, CatBoost run on multiple (4) threads on a CPU).
> - CatBoost throughput: `0.2kk examples/sec`
> - MLP throughput: `2.4kk examples/sec`
>
> Same caveats about implementation details and negligible difference in absolute numbers apply here: both models are fast at inference and could be sped up even further (using more threads, pruning trees, exporting NNs to TensorRT).
>
> We also provide all runtime information in the `report.json` files for every experiment in the submission source code (the link is provided in the main text).
>
> **No comparison to AutoML**
>
> When it comes to AutoML methods, we believe that the comparison is out of the scope of this study. The paper studies pretraining methods, which are generally applicable to deep learning models on tabular data. Thus, we compare models in a unified setup, studying only the pretraining part.
>
> AutoML methods, such as Autogluon operate on another level, jointly performing hyperparameter tuning, preprocessing, model selection, model stacking and ensembling over multiple methods. All methods we compare and propose in this paper could be integrated into the AutoML pipelines to boost their performance, but it requires additional work and research in the field of AutoML, we leave this for future work.
>
> **Clarifications**.
> - The percentage of the inputs are corrupted is a hyperparameter, which is tuned for all the methods (we have added clarifications to the experimental setup description)
> - There are multiple bolded results in tables 3 and 4 for the top-1 ranked models, which are not significantly different (e.g., `MLP`, `MLP-PLR` and `MLP-T-LR` with `mask+target` objective have rank one on the HI dataset in table 3: there is no statistically significant difference between them, so all three entries are bolded).
>
> We have also added the notebook we used for generating all the tables to the source code (see `bin/create_latex_tables_tukey.ipynb` via the same repository, provided in the paper). The `def find_best_algorithm(...)` function is used throughout the notebook to bold the entries in the tables.
>
> **Differences with Gorishniy et al. (2021)**
>
> The study [1] focuses on the model architectures, introduces multiple baselines: MLP, ResNet and FT-Transformer, and concludes that GBDTs substantially outperform NNs on some tabular tasks, but at the cost of increased training time.
>
> This work, however, builds on `[1]` and `[2]`, and studies pretraining as a means to increase the downstream performance of aforementioned models. The pretraining stage significantly boosts their performance.
>
> **Summary of the pretraining methods**.
>
> We've added a section with a table, summarizing all pretraining methods studied in this work, to the Appendix.
>
> **Minor issues**.
>
> Thanks for reporting on the issues! We've updated the draft with fixes.
>
> **References**
> - `[1]` Gorishniy, Yury, et al. "Revisiting deep learning models for tabular data." Advances in Neural Information Processing Systems 34 (2021): 18932-18943.
> - `[2]` Gorishniy, Yura, Ivan Rubachev, and Artem Babenko. "On Embeddings for Numerical Features in Tabular Deep Learning." arXiv preprint arXiv:2203.05556 (2022).

---

> > ### Comment · Reviewer_MXDS · 2022-11-16
> > **Response**
> >
> > I thank the authors for their response clarifying the majority of my concerns, and I have updated my score accordingly.

---

### Official Review · Reviewer_S291 · 2022-10-24

**Confidence:** 3
**Correctness:** 3
**Technical Novelty And Significance:** 2
**Empirical Novelty And Significance:** 2
**Recommendation:** 3

**Clarity, Quality, Novelty And Reproducibility:**

The paper is clearly written and is of decent quality.
However, the paper is not novel.
As the method is simpler, it should be pretty straightforward to reproduce.

**Strength And Weaknesses:**

Strengths:
- The paper generates some useful insights around how to pretrain a MLP model for tabular data and finetune the model to achieve stronger results on downstream tasks. There are also interesting discussions round pretraining objective. Self reconstruction and mask prediction are useful pretraining objectives, as compared to contrastive loss.

- Combing supervised loss and unsupervised loss in the pretraining helps further increase the downstream performance.

Weaknesses:
- The paper is not clear about its experimental setup: what is the pretraining dataset and what is the downstream datasets? Are they the same but with different losses?

- As the paper pointed out that adding supervised loss and unsupervised loss during pretraining enhances downstream task performance. But the paper does not provide any evidence if adding target task prediction loss during pretraining would benefit generalization? Whether adding the prediction of some tasks would hurt the performance on some other tasks with different loss functions?

- If there is not universal solution between self-prediction objective, when applied in real, how should we pick the optimal objective, given some mixture of downstream tasks?

**Summary Of The Paper:**

Deep learning has become an important method for tabular data. Even though pretraining based DL has been demonstrated successful in computer vision and NLP problems, it is challenging to make reliable conclusion about pretraining efficacy in tabular DL. This work provides a fully labeled tabular datasets to understand if pretraining helps tabular DL in a fully supervised setting and compare pretraining methods to the strong supervised baselines.

Using the target dataset, the method pretrain the model parameters with pretraining objectives, including generative positive pairs in contrastive-like objective or to corrupt the input for reconstruction in a self-prediction based objectives. In the second stage, the model is finetuned on the downstream classification or regression task.

**Summary Of The Review:**

The paper is insufficient for a publication in ICLR, as explained in the weakness section.

Please provide more detailed explanation on the experimental setup, particularly:
- The pretraining dataset, dataset size.
- Downstream tasks. tasks losses, labeled dataset size, etc.

Please also formulate the problem properly with the pretraining losses and downstream task prediction losses.

If the downstream tasks have different prediction losses, how should we pick the pretraining recipe?

---

> ### Author Response · Authors · 2022-11-14
> **Response to Reviewer S291**
>
> We thank the reviewer for providing useful feedback. We are glad that reviewer pointed out that our findings about self-supervised objectives and pretraining performance interesting. We point the reviewer to the general answer about experimental setup and discussion around it.
>
> **Are pretraining and downstream datasets the same?**
>
> We stress out that in tabular DL pretraining and downstream datasets are the same. The only difference is the objectives used and the percentage of the dataset used for finetuning. As we've discussed in the general response, in this work we focus on the setup, where the whole dataset is used during finetuning. So pretraining datasets, their properties, downstream tasks and their losses are all presented in the Table 1 of the paper.
>
> **Adding target task prediction loss benefits generalization?**
>
> We do show that adding downstream task prediction losses benefits generalization. See table 3 of the paper, `+ sup` variations of the pretraining losses are adding the downstream task prediction to the pretraining and result in better downstream performance -> better generalization. We generally do not observe that adding prediction hurts downstream performance on the downstream task (always no worse than training directly on the downstream task without preliminary pretraining)
>
> **How to pick the optimal objective**
>
> There is no universal solution, but we provide discussion on which objectives are useful and should be compared and tried in practice in the main takeaway of section 4.1.
>
> **The paper is not novel**.
>
> We respectfully disagree. The novelty of our submission is not in proposing a fancy new objective or method, but in a systematic study that reveals new important results about the efficacy of simple pretraining techniques that significantly improve the performance of tabular DL.

---

> > ### Comment · Reviewer_S291 · 2022-12-05
> > **Thanks for the response.**
> >
> > The reviewer is a bit confused, because when we talk about generalization, we usually mean generalizing to a different data distribution or a task where the task consumes a different dataset. If you pretrain with the same target data distribution, by adding supervised loss, it is going to overfit the dataset, while not being generalize to a different data distribution or a different downstream task. Please provide results on finetuning a different target distribution (downstream task).

---

> > > ### Author Response · Authors · 2022-12-07
> > > **Clarifications**
> > >
> > > We are sorry for the confusion, above, we initially interpreted the "generalization" term as "generalization to unseen test examples". If we treat "generalization" as "generalization to unseen data/target/task distributions" (e.g., as in `[1]`), then **this kind of generalization is out of scope for our work**. **Please, see the global response** in the main thread. To quickly reiterate, in our setup, we are dealing with a **fully labeled downstream dataset** and we question whether it is possible to achieve better metrics than with a simple supervised training. The answer is yes: by splitting the training into two stages (pretraining + finetuning; both stages use the same whole downstream dataset), and the paper focuses on the "pretraining" stage.
> > >
> > > `[1]` Roman Levin, Valeriia Cherepanova, Avi Schwarzschild, Arpit Bansal, C. Bayan Bruss, Tom Goldstein, Andrew Gordon Wilson, Micah Goldblum, "Transfer Learning with Deep Tabular Models"

---

### Official Review · Reviewer_QuBL · 2022-10-25

**Confidence:** 4
**Correctness:** 4
**Technical Novelty And Significance:** 3
**Empirical Novelty And Significance:** 3
**Recommendation:** 6

**Clarity, Quality, Novelty And Reproducibility:**

Although the novelty of the paper is a bit limited, the empirical results show some good practice for tabular data training. The experiments are convincing.


**Strength And Weaknesses:**

The paper compares various supervised pretraining objectives on tabular data. Some helpful conclusions are drawn from the experimental results.

Here are some suggestions:
1. The main assumption of the paper is that "Pretraining is always performed directly on the target dataset and does not exploit additional data", which is not always the case in the deep learning field. It seems the authors need to point out revisiting the "supervised" pretraining. The reviewer has two questions:
    1.1 There are some self-supervised learning strategies on tabular data, such as SubTab. Will they help get better performance?
    1.2 Could we pre-train on another dataset and then apply the pre-trained model to the target dataset?
2. The authors need to make the formulation of the pretraining and fine-tuning stages clear. Which learnable parameters are shared between the two stages? Some readers may not be familiar with the tabular model.
3. The pretraining objectives introduce a lot of augmentation strategies. Will they also help GBDT methods?
4. If we use different deep models like ResNet or FT-Transformer, will we get similar results?
5. The authors validate that pretraining is a good trick to boost the ability of deep tabular models. Does it mean that pretraining becomes a basic step when we use deep models on tabular data? Then how to compare various deep models in a fair manner? Different methods may need different pretraining strategies.

**Summary Of The Paper:**

This paper investigates the pretraining practice on tabular data. It shows that using the object target labels during the pretraining stage is beneficial for the downstream performance. Several target-aware pretraining objectives are proposed. Experiments on various datasets validate the claim.


**Summary Of The Review:**

The paper is clear and well-written. Several pretraining strategies are investigated in the paper. Experiments support the claim of the paper. There are some suggestions, and please see the weakness part.

---

> ### Author Response · Authors · 2022-11-14
> **Response to Reviewer QuBL**
>
> We thank the reviewer for their time, bringing up the important questions and suggesting the baselines. We answer the questions and provide additional experiments below.
>
>
> **Supervised assumption**
>
> We point the reviewer to the general answer above regarding the experimental setup and positioning. When it comes to the self-supervised learning strategies (such as SubTab), we compare most prominent variations of self-supervised pretraining paradigms: contrastive and self-prediction (SubTab is a variation of the self-prediction and contrastive objectives)
>
> **Pretraining/finetuning stages**
>
> We clarify that the backbone and embedding parameters are shared between pretraining on the dataset and finetuning on it, only the head is initialized randomly to finetune the model on the target task (regression or classification). See the pretraining section of the experimental setup for more details.
>
> **GBDTs with augmentations**
>
> We adopt the augmentation strategies to GBDTs by augmenting the datasets before GBDT trees construction, effectively enlarging datasets with random column resampling to match the number of pertrubations the neural network is exposed to during pretraining (tens of millions of randomly corrupted samples).
>
> Here is the example pseudocode producing the augmented dataset:
>
> ```python
> X = np.load('dataset.npy')
> N,M = X.shape
>
> N_aug = 10_000_000
> n_repeats = math.ceil(N_aug / N)
>
> X_clean = X.repeat(n_repeats, 1)[:N_aug]
> X_shuffle = np.take_along_axis(X, np.random.randint(0,N, size=(N_aug,M)), axis=0)
> X_aug = np.where(np.random.binomial(1,p=hyperparam_p), X_shuffle, X_clean)
> ```
>
> We show the results in the table below (the tuning protocol is the same for the `(aug)` version, except an additional resample probability parameter)
>
>
> | Method         | GE    | CH    | CA    | HO    | AD    | HI    |
> | -------------- |:----- |:----- |:----- |:----- |:----- |:----- |
> | Catboost       | 0.682 | 0.864 | 0.433 | 3.115 | 0.927 | 0.806 |
> | Catboost (aug) | 0.660 | 0.838 | 0.474 | 3.561 | 0.923 | 0.791 |
>
> We observe that adding noisy/augmented samples to the datasets deteriorates GBDTs performance significantly. We've also added this result in the appendix of the revision.
>
> **ResNet and FT-Transofmer**
>
> We do, in fact, evaluate pretraining for the FT-Transformer model and find that results are similar (see Appendix D of the paper).
>
> **Common practice?**
>
> We point out that pretraining in tabular DL can bring significant performance boosts for the state-of-the-art models. These boosts, however, come at a price of more computational resources, therefore, in resource-contrained settings, we recommend to use pretraining with caution. Otherwise, we suggest the practitioners to use pretraining for typical tabular problems.
>
> For research purposes, for instance, when comparing model architectures, it is better not to use pretraining, since it is "orthogonal" to model architecture improvements, and will make fair comparisons harder. As we observe in the experiments with FT-Transformer, basic MLPs, MLPs with various feature embeddings -- pretraining helps, regardles of the architecture.

---

> > ### Comment · Reviewer_QuBL · 2022-11-17
> > **Thanks for the response**
> >
> > Thanks for the detailed response. The general and specific responses resolve my concerns, especially the common practice part. The new title is clearer. My suggestion is that the authors also need to modify some parts of the main paper.

---

### Official Review · Reviewer_9GeL · 2022-10-29

**Confidence:** 4
**Correctness:** 3
**Technical Novelty And Significance:** 3
**Empirical Novelty And Significance:** 3
**Recommendation:** 8

**Clarity, Quality, Novelty And Reproducibility:**

Clarity: The paper is very clearly motivated and presented
Quality: High-quality systematic study on the effect of different pretraining techniques on deep learning for tabular data
Novelty: The paper's strength is not innovation, but this type of systematic study comparing traditional ML and deep learning for tabular data is very timely
Reproducibility: The paper's results seem reasonably reproducible thanks to the detailed description of the experiments and hyperparameters used

**Strength And Weaknesses:**

Strengths:
- Well-motivated and timely study that shows that deep learning models can consistently beat decision trees on tabular datasets
- Extensive study on multiple datasets and invesigating multiple pretraining procedures
- Interesting synthetic dataset construction to control feature relevance to study the effect of combine masked and label noise denoising pretraining
- Careful comparison of different types of MLPs and transformer models
- Useful study of computation and time cost due to pretraining

The paper has a few minor weaknesses, which can mostly addressed by clarifying some points.
- In he synthetic experiment, is importance rank in Fig. 1 given by the influence of the vector p? Or is that derived from the feature importances?
- During finetuning in the efficient ensembling case, is the whole architecture including the backbone trained end-to-end, or is just the heads that are being trained on a fixed backbone? This makes a great difference in terms of compute and memory: the difference between keeping around 15 different backbones and only one.
- The paper is rather complete in terms of citations of previous works, but is however missing some notably very connected papers. It would be for instance beneficial to cite the paper Padhi et al. "Tabular Transformers for Modeling Multivariate Time Series", ICASSP 2021, which also uses masked denoising pretraining (although in tabular time series as opposed to simple tabular datasets).

**Summary Of The Paper:**

This paper deals with tabular data, specifically examining in detail the question whether deep learning models can be trained to compete with traditional methods based on decision trees. In particular, the paper studies the role of pretraining, a training modality that is peculiar to deep learning and has been crucial in establishing its dominance in NLP and vision.
Through a extensive battery of experiments on several tabular datasets, the paper evaluates various pretraining methods and architectures, showing that specific types of pretraining allow deep learning models to be competitive against decision tree models on tabular dataset.
The most successful type of pretraining combines mask denoising and label noise, and the paper proposes an explanation of the success of label noise based on studying a synthetic dataset which allows to control the relevance of individual features to predict the target. This study reveals that masked and label noise denoising pretraining allows to better predict features that are less relevant to predict the target.
Finally, the paper shows that the same pretrained deep learning model can be finetuned multiple times, and all the multiple finetuned models combined to increase performance even further.

**Summary Of The Review:**

The paper presents a timely exhaustive study on the effect of pretraining for deep learning on tabular data, and how this can be used to beat traditional machine learning techniques like decision trees.

---

> ### Author Response · Authors · 2022-11-14
> **Response to Reviewer 9GeL**
>
> We thank the reviewer for the detailed review. We are glad that the reviewer found the study timely and recognized the strengths of our submission (we share the understanding that the systematic study brings value to the community and is an important contribution). We are also glad the reviewer found the synthetic data experiments insightful.
>
> We provide the clarifications below.
>
> - In the synthetic data experiments the importance rank in figure 1 is derived from the CatBoost feature importances. The probabilities are only used for dataset generation (we find that the CatBoost feature importances correlate well with these predefined probabilities). We've clarified this in the new revision.
> - In the efficient ensembling experiment, both the head and the backbone are finetuned. We agree that keeping 15 copies of the model makes a great difference, but the main gain here is in skipping the expensive (in terms of compute time) pretraining stage and creating the ensembles during much cheaper finetuning. No gain in terms of reducing the ensemble memory overhead is achieved through this.
> - We've added the citation for the paper to the discussion of pretraining for the tabular domain.

---

> > ### Comment · Reviewer_9GeL · 2022-11-17
> > **Acknowledgment of authors' response**
> >
> > I want to thank the authors for satisfactorily addressing the few outstanding questions raised in my comments.

---

### Author Response · Authors · 2022-11-14
**General response**

We thank all the reviewers for their time reviewing the paper and for providing useful feedback! We are glad that our study was found to be well-motivated and timely (`9GeL`), and the results to be helpful (`QuBL`) and useful (`S291`).

From the reviews, we see that our positioning, scope and experimental setup caused some confusion among the reviewers (e.g. see the reviews by `QuBL` and `S291`). We thank the reviewers for raising the questions. To address this, below, we:
1. clarify the scope and the experimental setup
2. propose minor changes to the title and the abstract to better reflect the scope of our work


## 1. Clarifying the scope and the experimental setup

**In many other papers** on pretraining (especially in computer vision and NLP), there are **two** "parts" of data:
- abundant *unlabeled* data, which is used for pretraining
- downstream *labeled* data, which is used for finetuning

All but `[1]` of the prior studies on pretraining in tabular DL imitate this setup by taking fully labeled datasets and discarding some of the labels.

By contrast, **in our paper**, we consider setups where we have only the second part (downstream labeled data). In this setup:

- the baseline solution is the vanilla supervised learning
- the only goal is to improve the performance on this downstream task (generalization to other downstream tasks is **NOT** a goal)
- the same downstream data is used for both pretraining and finetuning

In tabular data problems, this common practical scenario is underexplored. The examples of papers similar in spirit to our work are `[1]`, `[2]`, `[3]`.

## 2. Changes to the title and the abstract

To better reflect the scope described above, we propose the following changes to the title and the abstract. **Importantly**, we only update the wording and terminology to make the paper clearer, everything else (scope, experiments, related work etc.) are not affected in any way.

**New title**: Pretrain then Finetune: Getting the Most from Your Labeled Dataset in Tabular Deep Learning

**New abstract**: Similarly to computer vision and NLP, deep learning models for tabular data can benefit from pretraining, which is not the case for traditional ML models, such as gradient boosted decision trees (GBDT). Although the pretraining techniques for tabular data are actively studied, the existing works mostly focus on the unsupervised pretraining, implying the access to a large amount of unlabeled data in addition to the labeled target dataset. By contrast, pretraining in the fully supervised setting, when the available data is fully labeled and directly represents the downstream tabular task, receives significantly less attention. Moreover, the existing works on pretraining typically consider only the simplest MLP architectures and do not cover the recently proposed tabular models.

In this work, we aim to identify the best practices for pretraining in tabular DL that can be universally applied to different datasets and architectures in the fully supervised setting. Among our findings, we show that using the object target labels during the pretraining stage is beneficial for the downstream performance and advocate several target-aware pretraining objectives. Overall, our experiments demonstrate that properly performed pretraining significantly increases the performance of tabular DL models on fully supervised problems.

**Other changes**: there are also minor changes to the text that additionally highlight our scope and positioning.

**References**

- `[1]` Bahri, Dara, et al. "Scarf: Self-supervised contrastive learning using random feature corruption." arXiv preprint arXiv:2106.15147 (2021).
- `[2]` El-Nouby, Alaaeldin, et al. "Are Large-scale Datasets Necessary for Self-Supervised Pre-training?." arXiv preprint arXiv:2112.10740 (2021).
- `[3]` Krishna, Kundan, et al. "Downstream Datasets Make Surprisingly Good Pretraining Corpora." arXiv preprint arXiv:2209.14389 (2022).

---

### Decision · Program_Chairs · 2023-01-20

**Decision:**

Reject

**Justification For Why Not Higher Score:**

AC also carefully read this paper, and generally agrees with the concerns of reviewers. AC found that the idea of pretraining on the same dataset is well-known and the improvements look marginal. Hence, AC thinks the observations of this paper are somewhat straightforward (even though the tabular domain is new). Moreover, AC feels this paper is not ready for publication at this moment due to the unsatisfactory presentation quality throughout this paper. For example, as many reviewers pointed out, there is no clear description and formulation about pretraining objectives. AC strongly suggests revising the manuscript with a more neat, clear, and comprehensive presentation. Considering all aspects, AC thinks this paper does not meet the high standard of the ICLR community, so AC tends to recommend rejection.

**Justification For Why Not Lower Score:**

N/A

**Metareview: Summary, Strengths And Weaknesses:**

This paper studies the effect of pretraining on the tabular benchmarks. Specifically, the authors focus on fully-supervised setups without any extra (unlabeled) dataset, and they examine various pretraining objectives on fully-labeled datasets by pretraining and fine-tuning on the same dataset. At the initial reviews, this paper received both strong positive and negative scores (8 vs 3), and the reviewers did not reach an agreement during the discussion period. The reviewers' concerns are the limited novelty, the marginal improvements, and the unclear presentation, although the paper provides some useful insights around how to pretrain a MLP model for tabular data.